# Overcoming adaptive resistance to anti-VEGF therapy by targeting CD5L

Christopher J. LaFargue[1,23], Paola Amero [2,3,23], Kyunghee Noh[1,4,23], Lingegowda S. Mangala[1,5], Yunfei Wen [1] ✉, Emine Bayraktar[1], Sujanitha Umamaheswaran [1], Elaine Stur[1], Santosh K. Dasari[1], Cristina Ivan[2], Sunila Pradeep[6], Wonbeak Yoo [7], Chunhua Lu[1], Nicholas B. Jennings[1], Vinod Vathipadiekal[8,9], Wei Hu[1], Anca Chelariu-Raicu[1,10,11], Zhiqiang Ku [12], Hui Deng[12], Wei Xiong[12], Hyun-Jin Choi[13,14], Min Hu[15], Takae Kiyama[16], Chai-An Mao[16,17], Rouba Ali-Fehmi[18], Michael J. Birrer[19], Jinsong Liu [20], Ningyan Zhang [12], Gabriel Lopez-Berestein[2,5], Vittorio de Franciscis[21,22], Zhiqiang An [12] & Anil K. Sood [1,5] ✉

Antiangiogenic treatment targeting the vascular endothelial growth factor (VEGF) pathway is a powerful tool to combat tumor growth and progression; however, drug resistance frequently emerges. We identify CD5L (CD5 antigen-like precursor) as an important gene upregulated in response to anti-angiogenic therapy leading to the emergence of adaptive resistance. By using both an RNA-aptamer and a monoclonal antibody targeting CD5L, we are able to abate the pro-angiogenic effects of CD5L overexpression in both in vitro and in vivo settings. In addition, we find that increased expression of vascular CD5L in cancer patients is associated with bevacizumab resistance and worse overall survival. These findings implicate CD5L as an important factor in adaptive resistance to antiangiogenic therapy and suggest that modalities to target CD5L have potentially important clinical utility.

Angiogenesis is well known to play an important role in tumor development and growth[1]. This complex process relies on the careful orchestration of many factors, including vascular endothelial growth factor (VEGF) and its receptor (VEGFR), fibroblast growth factor (FGF), and others[2]. Many antiangiogenic drugs, particularly those targeting the VEGF/VEGFR pathway, have been developed and are approved for cancer treatment. Although many patients benefit from such therapies, virtually all patients eventually experience relapse or progression of the disease. Understanding and overcoming adaptive changes to anti-VEGF drugs represent an opportunity to further enhance the efficacy of these drugs and potentially delay or prevent adaptive resistance[3–6].

To examine potential mechanisms underlying resistance to anti-VEGF antibody (AVA) therapy, we used mouse models to identify tumors that demonstrated growth subsequent to a period of initial response to treatment. Specifically, we established orthotopic mouse models of ovarian cancer designed to develop adaptive resistance after treatment with an AVA. We examined the genomic profiles of tumor-associated endothelial cells collected at pretreatment, at the maximal response, and at tumor progression and found substantially elevated CD5L levels at the time of progression. CD5L, also known as apoptosis inhibitor expressed by macrophages (AIM), was previously identified as a soluble protein secreted primarily from macrophages in lymphoid tissues during inflammatory response[7]. Additional roles of CD5L have been discovered since, but those related specifically to endothelial cells and angiogenesis remain unknown. Here, we present data implicating CD5L involvement in adaptive resistance to bevacizumab. We also demonstrate that neutralizing CD5L by using an antibody or aptamer blocked adaptive resistance to anti-angiogenic therapy. Anti-CD5L drugs could potentially be used to overcome resistance to bevacizumab and other antiangiogenic therapies.

## Results

### Adaptive genomic changes in tumor endothelial cells

To identify possible targets involved in adaptive resistance, we used the SKOV3ip1 ovarian cancer mouse model. Mice were treated with the B20 anti-VEGF antibody, and tumors were obtained at various time points that demonstrated either sensitivity or resistance (Fig. 1A). Endothelial cells were then isolated from sensitive and resistant tumor samples, and gene expression profiling was performed by using isolated mRNA. A large number of genes displayed differential expression between the endothelial cells from sensitive *versus* resistant tumors, with *CD5L* demonstrating the largest difference: 27.48-fold higher in the resistant endothelial cells (Fig. 1B). Immunohistochemical analysis showed that CD5L expression in endothelial cells from resistant tumors was significantly higher than in endothelial cells from sensitive tumors (Fig. 1C). To determine the expression of CD5L in other tumor cell types, we analyzed five high-grade serous ovarian cancer samples by using single-cell RNA sequencing of six populations including T cells, monocytes, epithelial cells, fibroblasts, natural killer cells, and B cells. Fig. S1 shows the UMAP of single cell data by sample (A) and by cell type (B). We observed almost no expression of CD5L in any of these populations, with only a few monocytes and B cells showing some level of expression (each dot represents one single cell; Fig. S1C, D).

Next, we examined the biological effects of CD5L upregulation in tumor endothelial cells. To determine the function of CD5L in tumor angiogenesis, we generated CD5L-overexpressing RF24 endothelial cells (Fig. 1D). These cells displayed increased proliferation, tube-formation capacity, and cell migration compared with control cells (Fig. 1E–G). Consistent with CD5L being a primarily secreted protein, we found that the concentration of CD5L in the conditioned media from RF24 cells overexpressing CD5L was significantly higher than the CD5L concentration in the media from control RF24 cells (empty vector) (Fig. 1H). To confirm that overexpression of CD5L was the primary source of these observed effects, we next treated control RF24 cells with *CD5L* siRNA. More than a 90% knockdown of CD5L protein levels was seen within 72 h compared with a non-targeting siRNA (Fig. 1I). Notably, cells treated with *CD5L* siRNA showed reduced proliferation, tube-formation capacity, and cell migration compared with cells treated with control siRNA (Fig. 1J–L). Furthermore, we used CRISPR/Cas9 to knock out *CD5L* completely in RF24 endothelial cells, and the *CD5L* CRISPR/Cas9 knockout cells formed fewer tubes than the scramble control cells. Importantly, the addition of CD5L recombinant protein to these cells rescued the decreased tube formation induced by *CD5L* knockout (Fig. S2A, B).

### CD5L is upregulated through hypoxia-induced PPARG overexpression

To determine possible mechanisms of CD5L elevation in tumor endothelial cells, we next examined the regulation of *CD5L* gene transcription. Upon analysis of the *CD5L* promoter sequence, we identified a putative binding site (shown in red, Fig. S3) for the transcription factor PPARG. To determine whether PPARG may serve as an upstream regulator of CD5L, we ectopically expressed PPARG in RF24 endothelial cells. Compared with controls, endothelial cells with elevated PPARG demonstrated a significant increase in both CD5L mRNA and protein (Fig. 2A, B).

We also generated a *CD5L* promoter construct (pCD5L WT) and found a significant increase in luciferase activity when transfected into PPARG overexpressing RF24 cells compared with wild-type RF24 cells (Fig. 2C). To further prove that PPARG expression was responsible for the observed increase in CD5L expression and promoter activity, we next treated wild-type RF24 cells with *PPARG* siRNA. Cells treated with si*PPARG* showed a reduction in both *CD5L* mRNA and protein (Fig. 2D, E), as well as significantly decreased luciferase activity of the *CD5L* promoter construct (Fig. 2F). We next deleted the *PPARG* binding site from the *CD5L* promoter construct (pCD5L del) to

determine whether the observed CD5L increase was due to PPARG specific binding. After co-transfecting RF24 cells with PPARG expression plasmid and either p*CD5L* WT or p*CD5L* del, we found that the mutated PPARG binding site resulted in significantly reduced luciferase activity compared with the non-mutated promoter, indicating that PPARG directly regulates CD5L expression (Fig. 2G).

Next, we sought to determine whether any other factors upstream of PPARG played a role in the upregulation of CD5L. With the use of our initial gene expression dataset generated from AVA-resistant endothelial cells, we performed an ingenuity pathway analysis and found a close correlation with hypoxia signaling proteins (HIF1α, EPAS1, and ARNT) (Fig. S4).

Gene expression profiling of B20-sensitive and -resistant tumor endothelial cells also showed that both *PPARG* and *HIF1α* expression levels were higher (the difference was significant for *HIF1α*) in resistant endothelial cells compared with sensitive endothelial cells (Fig. S5A). To confirm this relationship, we grew RF24 cells under hypoxic and normoxic conditions; in cells grown in hypoxic conditions, there was a significant increase in both *PPARG* and *CD5L* mRNA and protein levels compared with levels grown in normoxic conditions (Fig. 2H, I). Extending this finding further, we incubated RF24 cells with a HIF1α stabilizing compound (cobalt chloride, $CoCl_2$) for 6 and 30 h under normoxic conditions. We found that both *PPARG* and *CD5L* mRNA expression levels were significantly higher after 30 h of $CoCl_2$ incubation than after 6 h as well as non-treated cells, further validating that hypoxia-like conditions lead to increased *PPARG* and *CD5L* expression (Fig. 2J). In addition, we found that CD5L and HIF1α protein expression levels were also increased at longer incubation times of $CoCl_2$ (Fig. 2K). To determine whether HIF1α blockade would result in the opposite effect on PPARG and CD5L, we used two known HIF1α inhibitors, YC-1 and topotecan, under hypoxic conditions. After treatment of RF24 cells with either YC-1 or topotecan in hypoxic conditions, we found a significant decrease in both *PPARG* and *CD5L* mRNA expression compared with levels in control cells treated with DMSO (Fig. 2L). In addition, *HIF1α* siRNA-treated cells showed decreased *PPARG* and *CD5L* mRNA expression (Fig. S5B). Furthermore, we found that *CD5L* promoter activity (p*CD5L* WT) was significantly increased under hypoxic *versus* normoxic conditions (Fig. 2M).

Finally, we performed chromatin immunoprecipitation (ChIP) analysis of the *CD5L* promoter by using an anti-PPARG antibody under hypoxic and normoxic conditions. We found that the PPARG binding site (located in region 1 of the *CD5L* promoter) had significantly higher fold-enrichment for PPARG under hypoxic than under normoxic conditions (Fig. 2N). Moreover, we demonstrated the selectivity of PPARG for the specific promoter sequence in region 1 since only minimal binding was observed in regions 2 or 3 under normoxic or hypoxic conditions.

### Exogenous CD5L increases PI3K/AKT signaling in endothelial cells

Since CD5L is primarily a secreted protein shown to act in a paracrine fashion, we next exogenously treated RF24 endothelial cells with CD5L to determine the downstream signaling effects. We performed reverse phase protein array (RPPA) analyses of both CD5L-treated and untreated RF24 endothelial cells and found that the CD5L-treated cells showed activation of PI3K/AKT signaling (Fig. S6A). To validate these results, we measured the protein levels of AKT and phospho-AKT in CD5L-treated RF24 cells compared with untreated controls. The RF24 cells, which were pretreated with CD5L, had increased expression of pAKT compared with untreated cells (Fig. 3A). Furthermore, the addition of the PI3K inhibitor LY294002 after CD5L pretreatment mitigated these stimulating effects on pAKT (Fig. 3B). Similarly, both tube formation and cell migration of RF24 cells increased significantly after CD5L treatment (Fig. S6B, C); however, both were significantly decreased with the co-addition of a PI3K inhibitor (Fig. 3C, D). Since

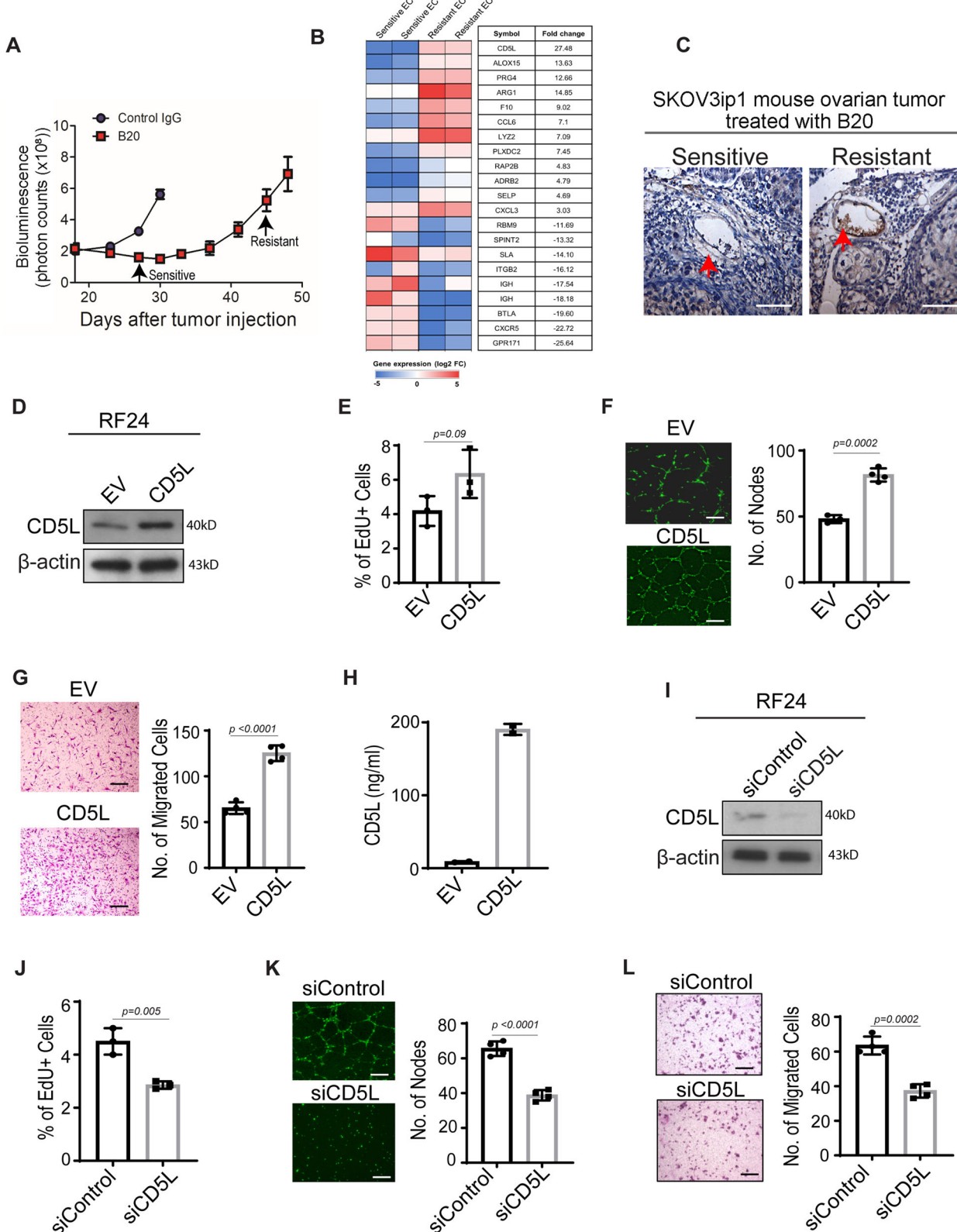

CD36 has been previously reported to be one of the major receptors for CD5L in macrophages[7,8], we sought to determine the effect of exogenous CD5L treatment on CD36 expression. CD5L-treated RF24 cells had higher expression of *CD36* than did untreated RF24 cells (Fig. 3E), possibly through a positive-feedback mechanism. To determine whether CD36 was responsible for the CD5L-dependent upregulation of PI3K/AKT, we transfected RF24 cells with *CD36* siRNA and again exogenously treated them with CD5L. Importantly, knockdown of CD36 in RF24 cells decreased pAKT level even with CD5L treatment (Fig. 3F). RF24 cells exogenously treated with CD5L, compared with control cells, have reduced sensitivity to bevacizumab (Fig. 3G). Conversely, treatment of RF24 cells with *CD5L* siRNA, compared with siControl treated cells, resulted in enhanced sensitivity to bevacizumab (Fig. 3H).

**Fig. 1 | Upregulation of CD5L in anti-VEGF therapy–resistant endothelial cells promotes angiogenesis properties. A** Time point at which SKOV3ip1 ovarian cancer tumors were isolated during the course of B20 treatment. Tumor progression was identified by an increase in bioluminescence (Data represented as mean ± SD; $n = 5$ mice for control IgG and $n = 10$ mice for B20 antibody treatment). **B** Heat map from gene expression profiling of endothelial cells isolated from B20-resistant tumors compared with endothelial cells isolated from B20-sensitive tumors. The microarray data were deposited in GEO (Accession number GSE180687). **C** CD5L staining in endothelial cells from mouse tumors sensitive or resistant to B20 antibody ($n = 4$ mice; scale bar = 100 μm). **D** CD5L protein expression in RF24 endothelial cells containing CD5L-overexpressing plasmid versus empty vector (EV). Western blotting was performed two times as technical replicates; in each repeat, the blotting, including loading control, was performed using the same sample processing controls. **E** Cell proliferation in RF24 endothelial cells containing CD5L-overexpressing plasmid versus EV. **F, G** Tube formation (**F**) and cell migration (**G**) in RF24 endothelial cells containing CD5L-overexpressing plasmid versus EV (scale bar = 200 μm). **H** Concentration of CD5L in media collected from RF24 endothelial cells containing CD5L-overexpressing plasmid versus EV ($n = 2$ biologically independent experiments). **I** Levels of CD5L protein in RF24 endothelial cells treated with si*CD5L* versus siControl. Western blotting was performed two times as technical replicates; in each repeat, the blotting, including loading control, was performed using the same sample processing controls. **J–L** Cell proliferation (**J**), tube formation (**K**), and cell migration (**L**) in RF24 cells treated with siCD5L versus siControl; (scale bar = 200 μm for **K** and **L**). Data represented as mean ± SD, determined by two-tailed Student's $t$-test; $n = 3$ for **E**, **F**, and **J** and $n = 4$ for **G**, **K**, and **L** biologically independent experiments.

## Silencing of PPARG inhibits angiogenesis and tumor growth

Next, to investigate the role of PPARG in regulating CD5L downstream effects, we generated a knockout of *PPARG* in RF24 endothelial cells with the use of CRISPR/Cas9 and tested its effects on CD5L-induced angiogenic properties such as tube formation. Results indicated that CRISPR knockout of *PPARG* led to decreased tube formation compared with scramble control-treated cells. Importantly, the addition of CD5L recombinant protein to these cells rescued the decreased tube formation under *PPARG* knockout (Fig. S7).

Next, to further explore the function of PPARG and CD5L in tumor endothelial cells, we injected murine ID8 ovarian cancer cells into the peritoneal cavity of C57BL/6 mice containing an endothelial cell–specific *PPARG* knockout or into C57BL/6 WT mice (Fig. 4A)[9]. We observed a 50% reduction in tumor weight and in the number of tumor nodules in the *PPARG* KO mice compared with WT mice (Fig. 4B, C). Moreover, immunohistochemical (IHC) analysis of tumor tissues from *PPARG* KO versus WT mice revealed a significant decrease in cell proliferation and microvessel density in the *PPARG* KO mice (Fig. 4D, E).

We next performed a survival analysis of *PPARG* KO mice versus WT mice with concurrent anti-VEGF treatment. *PPARG* KO mice had significantly improved survival while receiving the B20 anti-VEGF antibody treatment, compared with the WT mice (Fig. 4F). Consistent with our in vitro data, we found that tumor samples from *PPARG* KO mice had lower pAKT expression than did tumors from WT mice (Fig. 4G).

## Antibodies targeting CD5L exhibit antitumor and anti-angiogenic effects

Considering our finding that AVA resistance is mediated, in part, by overexpression of CD5L, we next aimed to develop an antibody to specifically target CD5L. We generated a large panel (>350 binding hits) of anti-CD5L monoclonal antibodies by using two strategies: (1) screening single B cells isolated from CD5L antigen–immunized rabbits and (2) panning human antibody phage display libraries. We selected ten antibodies for further evaluation based on in vitro characterization of binding affinity (Kd), CD5L mouse cross-reactivity, and binding epitopes. We then screened these ten antibodies in vivo with the use of an orthotopic ovarian cancer mouse xenograft tumor model (SKOV3ip1). Among those tested in vivo, two monoclonal antibodies (H-447 and R-35) showed significant tumor reduction when compared with the isotype antibody control (Fig. 5A–C). Mice treated with the isotype control antibody had a larger tumor burden than did mice treated with an effective anti-CD5L antibody (Fig. 5A). Treatment with the two effective anti-CD5L antibodies resulted in significantly lower tumor weights and fewer tumor nodules compared with the control antibody (Fig. 5B, C). Treatment with the R-35 antibody resulted in significantly fewer tumor blood vessels than did treatment with the control antibody (Fig. 5D). Similarly, the addition of the R-35 antibody to RF24 cells treated with CD5L protein negated the observed increase in tube formation and cell migration observed when RF24 cells were treated with control antibody plus CD5L protein alone (Fig. 5E, F).

We also tested the effect of the R-35 antibody on primary endothelial cells. As shown in Fig. S8A, B, human pulmonary artery endothelial cells (HPAECs; Fig. S8A) and human umbilical venous endothelial cells (HUVECs; Fig. S8B) showed increased tube formation when treated with CD5L recombinant protein; this effect was significantly reduced in the presence of the R-35 antibody. Similarly, treatment of HUVECs with CD5L recombinant protein resulted in increased capillary formation and sprout length. The addition of the R-35 antibody to HUVEC spheroids treated with CD5L protein blocked the observed increase in sprout length in HUVEC spheroids induced by the control antibody plus CD5L protein (Fig. S8C).

Prior to carrying out experiments to assess the potential toxicity of the CD5L antibody, we checked the expression of CD5L in normal organ vasculature, including hepatic arterial, portal venous (PV), continuous, discontinuous, and capillary endothelial cells. As shown in Fig. S9A, these areas showed weak to modest CD5L expression. Next, we examined the effects of CD5L (R-35) antibody on normal C57BL/6 mice. Animals were treated with either control Ab or R-35 for 2 weeks (once per week). Blood was collected, and a complete blood count and other analyses were performed. As shown in Fig. S9B, no significant difference was observed in white blood cell count, hemoglobin level, or platelet count. Moreover, there were no differences in serum alanine aminotransferase (ALT), aspartate aminotransferase (AST), and lactate dehydrogenase (LDH) levels between R-35 and control Ab-treated mice. H&E staining of multiple organs, including lung, liver, kidney, and spleen, from R-35 treated mice, showed no differences in histopathological findings compared with control Ab-treated animals (Fig. S9C). To determine how the R-35 antibody affects endothelial cells in non-tumor models, we stained blood vessels of various organs by using a CD31 antibody and observed no differences between the control Ab-treated and R-35 Ab-treated groups, suggesting that CD5L blocking antibodies did not affect angiogenesis in normal organs (Fig. S9D).

To elucidate the downstream effects of blocking endothelial CD36-mediated CD5L signaling, we obtained an endothelial-specific CD36*flox/flox* knockout mouse model CD36*f/f*Tie2*Cre* (Fig. S10) and inoculated the mice with syngeneic murine ovarian cancer ID8-Luc cells[10]. In this model, the anti-CD5L antibody (R-35) had no significant effect on tumor growth when compared with its effect on WT C57BL/6 mice. Next, we checked the effects of the R-35 antibody on angiogenesis in endothelial-specific *CD36* knockout tumors by staining the tumor tissue with pAKT and CD31 antibodies. No difference was observed between *CD36* endothelial-specific KO mice treated with either R-35 antibody or IgG compared with mice treated with WT-R-35 (Fig. S11).

It has been reported that altered endothelial CD36 can change fatty acid uptake in endothelial cells[10]. Therefore, we tested exogenous fatty acid uptake capacity in RF24 cells with or without *CD36* knockdown. CD36 knockdown reversed the enhancement of fatty acid uptake induced by CD5L protein in endothelial cells (rCD5L; Fig. S12).

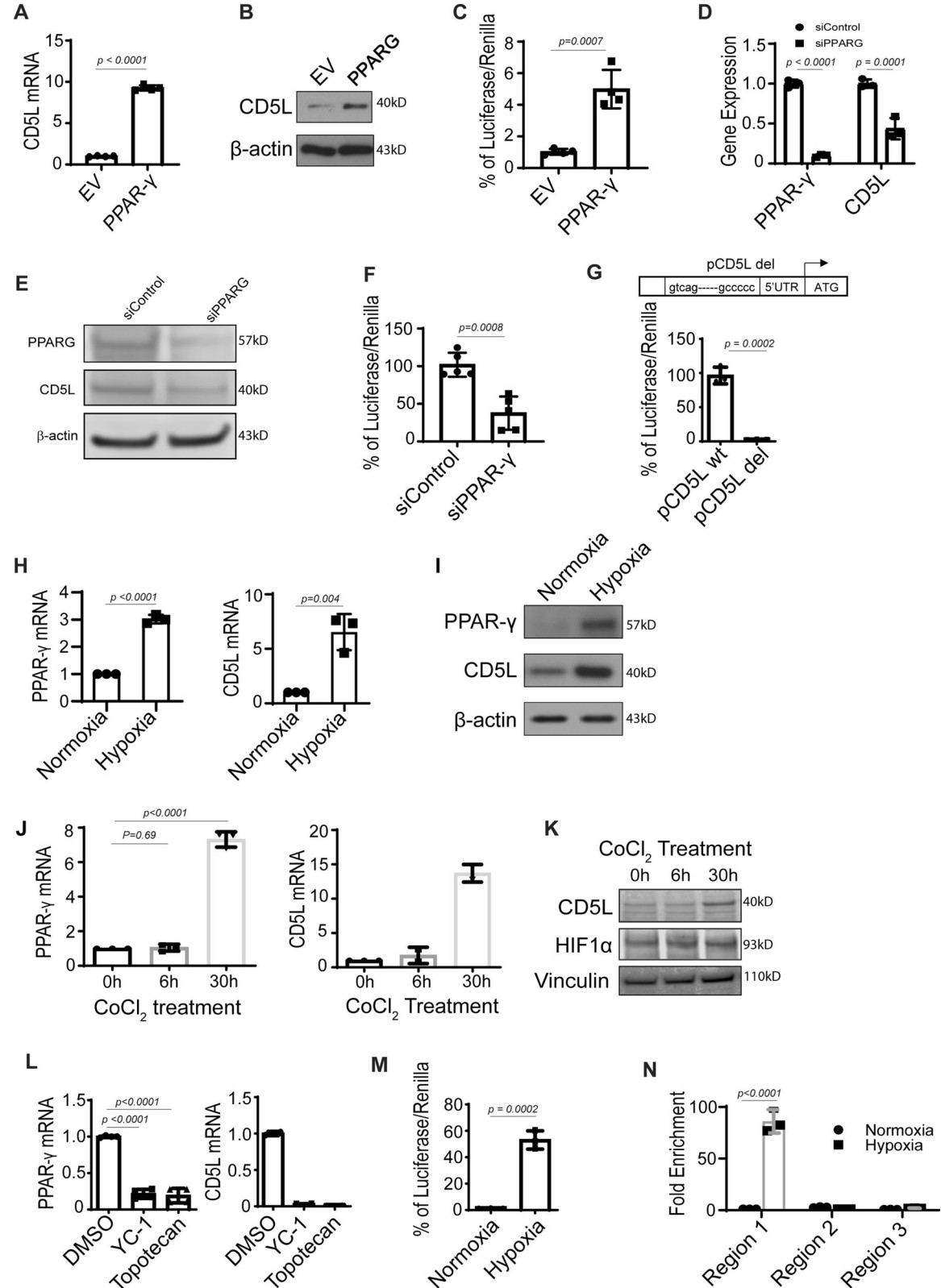

**S76.T (CD5L)-aptamer blocks resistance to anti-VEGF therapy**
To further assess the role of blocking CD5L in overcoming resistance to anti-VEGF therapy, we used an alternative strategy using aptamers. We selected the S76.T RNA-aptamer, which is a 2′-fluoro pyrimidine RNA-aptamer targeting CD5L. This sequence was selected by adopting a tandem variant of the protein–Systematic Evolution of Ligands by Exponential Enrichment (SELEX) procedure (Fig. S13). Upon the first

SELEX, we identified four sequences (S5, S47, S63, and S72) from the phylogenetic tree that could potentially bind recombinant CD5L (Fig. S14A). Among these sequences, the best aptamer candidate for CD5L was the S5 sequence with a Kd of 555.8 ± 148.9 nM (Fig. S14B), determined by microscale thermophoresis. Indeed, S63 and S72 sequences had no or weak binding for CD5L, whereas S47 was able to bind VEGF (used as a negative control), and CD5L target protein

**Fig. 2 | CD5L is upregulated through hypoxia-induced PPARG overexpression.**
**A, B** *CD5L* mRNA (**A**) and protein expression (**B**) in RF24 endothelial cells containing *PPARG* overexpressing plasmid versus empty vector (EV). **C** *CD5L* promoter constructs activation using RF24 endothelial cells containing *PPARG* overexpressing plasmid versus EV. **D, E** *PPARG* and *CD5L* mRNA (**D**) and protein (**E**) expression in RF24 endothelial cells treated with si*PPARG* versus siControl. **F** *CD5L* promoter construct activation using RF24 endothelial cells treated with si*PPARG* versus siControl. **G** Luciferase expression in RF24 endothelial cells after co-transfection of *PPARG* overexpressing plasmid and *CD5L* promoter construct harboring mutated *PPARG* binding site. **H, I** *PPARG* and *CD5L* mRNA (**H**) and protein (**I**) expression in RF24 endothelial cells cultured in hypoxic or normoxic conditions. **J, K** *PPARG* and *CD5L* mRNA (**J**) and protein (**K**) expression in RF24 endothelial cells treated for 6 and 30 h with cobalt chloride (HIF1α stabilizer). Western blots were performed from two independent technical replicates; in each repeat, the blotting, including loading control, was performed using the same sample processing controls (**B, E, I, K**); **L** *PPARG* and *CD5L* mRNA expression in RF24 endothelial cells treated with YC-1 or topotecan under hypoxic conditions. **M** *CD5L* WT promoter construct activation in RF24 endothelial cells cultured in hypoxic and normoxic conditions. **N** Chromatin immunoprecipitation (ChIP) analysis of the *CD5L* promoter using an anti-PPARG antibody under hypoxic and normoxic conditions. Data represented as mean values ± SD, determined by two-tailed Student's *t*-test for **A, C, F, G, H, M**; one-way ANOVA Tukey's multiple comparisons for **J** and **L**; two-way Anova Tukey's multiple comparisons for **D** and **N**; *n* = 3 for **D, G, H, J** left panel, **M** and **N**; *n* = 4 for **A, C,** and **L** left panel; *n* = 5 for **F**; *n* = 2 for **J** right panel and **L** right panel.

(Table S1). The most stable predicted secondary structure of S5 full-length, obtained by RNA structure prediction software, showed three hairpin-like structures followed by single-strand ends (Fig. S14C).

To identify a sequence with higher affinity binding for CD5L, using the mutagenized S5 aptamer as the starting pool, we performed the second selection introducing two more rounds of protein-SELEX (Fig. S13). Through the second SELEX, we identified S11, S23, S29, and S76 from the phylogenetic tree (Fig. S15A). S11, S23, S29, and S76 sequences showed points of insertion or deletion compared with the starting sequence S5 (Fig. S15B). Among all of the sequences, S76 was the aptamer with the best $K_d$ value (2.2 ± 0.7 nM) (Table S2).

Next, to determine the functional part of the aptamer involved in the recognition of the CD5L target, we analyzed the secondary structures and identified a common hairpin-like structure highlighted by a red rectangle (Fig. S16A–E). After the truncation of the S76 sequence, named S76.T, we analyzed the binding affinity to evaluate whether the truncated version could bind CD5L with the same or better affinity (Fig. S17A–D). S76.T had a Kd of 10.1 ± 2.3 nM, which is close to the $K_d$ value of the full-length sequence S76.

To assess the functional effects of the S76.T-aptamer, we used AVA (bevacizumab)-resistant RF24 cells. We evaluated pAKT expression in AVA-resistant cells after treatment with S76.T. The S76.T-aptamer reduced the expression of pAKT (Fig. S18A) and significantly inhibited tube formation and cell migration (Fig. S18B, C) compared with the scrambled aptamer in AVA-resistant cells. We treated AVA-resistant RF24 cells with S76.T alone or in combination with B20 (anti-VEGF antibody). Combined treatment significantly reduced the viability of AVA-resistant RF24 endothelial cells compared with either treatment alone (Fig. S18D).

To determine the effects of S76.T in adaptive resistance in vivo, we injected SKOV3ip1 ovarian cancer cells into the peritoneal cavity of nude mice. To generate an adaptive resistance tumor model, we first treated tumor-bearing mice with B20 until tumor growth was noted (~4 weeks). Next, we combined B20 treatment with S76.T injected intravenously (Fig. 6A). Mice treated with aptamer alone showed reduced tumor burden, whereas mice treated with the combination of aptamer and B20 showed a greater reduction in tumor weight, fewer tumor nodules (Fig. 6B–D), and lower microvessel density (Fig. 6E) and proliferation (Fig. 6F) compared with scramble IgG-treated mice.

**CD5L overexpression is associated with bevacizumab resistance and worse overall survival in ovarian cancer patients**

To determine the potential clinical relevance of CD5L overexpression, we first interrogated a select cohort of ovarian cancer patients identified as bevacizumab responders versus non-responders. We found that patients with disease resistant to bevacizumab had significantly higher CD5L expression in their tumor endothelial cells than did those with bevacizumab-sensitive disease (Fig. 7A). Furthermore, we found that patients with bevacizumab-resistant disease had significantly higher serum CD5L levels than did patients with bevacizumab-sensitive disease (Fig. 7B). In addition, we performed IHC staining for CD5L on a tissue microarray (TMA) consisting of tumor samples from an ovarian cancer cohort. CD5L expression in tumor endothelial cells ranged from absent or low (Fig. 7C, upper) to high (Fig. 7C, lower) in this cohort. In patients with high-grade serous ovarian cancer, those with high expression of tumor endothelial CD5L had significantly worse overall survival than those with low expression (Fig. 7D).

## Discussion

Our findings showed that CD5L is an important mediator of AVA resistance. From a conceptual perspective, we concluded that hypoxia incurred due to prolonged VEGF blockade ultimately drives the overexpression of CD5L through the upregulation of transcription factor PPARG. Analysis of the downstream pathways demonstrated prominent activation of the PI3K/AKT pathway with increased CD5L signaling in tumor endothelial cells (Fig. 8). Importantly, blocking CD5L with the use of a function-blocking antibody or RNA-aptamer restored the response to anti-VEGF therapy.

Anti-VEGF drugs have been approved for the treatment of many different cancer types. Unfortunately, although initial response rates have been high with these therapies, most patients develop resistant disease within weeks to months. The mechanisms underlying such adaptive resistance are likely to be multifactorial and are not fully understood. One such mechanism involves the upregulation of pro-angiogenic factors other than VEGF in response to AVA treatment, such as fibroblast growth factor 1 and ephrin A1[11]. In addition, tumors treated with AVAs have been shown to develop adaptive resistance to treatment by increasing their invasive potential without relying on new vessel generation[12,13]. Regardless of the exact mechanism, the phenomenon of tumor endothelial cells adapting to their specific microenvironment is now well accepted. Therefore, we took a systematic approach whereby we performed mRNA profiling on tumor endothelial cells obtained from mouse models with adaptive resistance to AVA and identified CD5L as the most highly upregulated mRNA in treatment-resistant endothelial cells.

Initially named for its anti-apoptotic role in leukocytes[14], CD5L has since been implicated as an important regulator of inflammatory responses, particularly through its effect on macrophages. It has also been shown to be involved in a variety of cellular processes, including atherosclerosis, infection, and cancer[7]. Although CD5L is secreted primarily by macrophages, it has been shown to have diverse roles in the immune system. Mice that are deficient in CD5L have reduced lymphocytes in liver granulomas when challenged with heat-killed *C. parvum* compared with levels in wild-type mice[15]. In addition, in vitro studies using liver-associated T cells and natural killer T (NKT) cells from mice exposed to *C. parvum* showed significant inhibition of apoptosis after treatment with recombinant CD5L[15]. CD5L has also been shown to induce the formation of bronchoalveolar adenocarcinoma in a transgenic mouse model with CD5L-overexpressing myeloid cells[16]. Interestingly, CD5L seems to have a protective role in mouse hepatocellular carcinoma through its interaction with CD55, CD59, and Crry, leading to subsequent complement activation and induced necrotic death of hepatocytes[17]. The scavenger receptor CD36 has

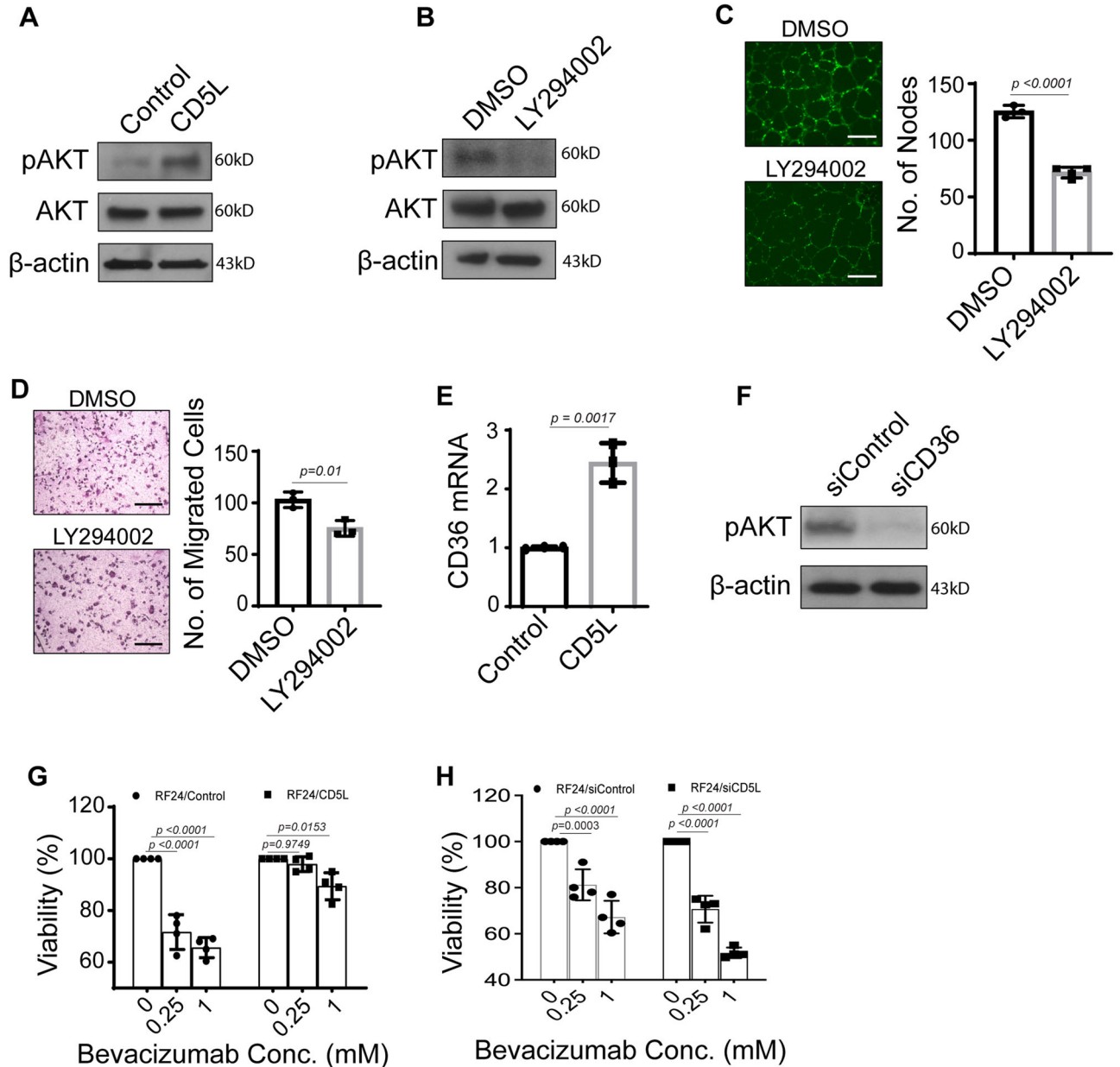

**Fig. 3 | Exogenous CD5L treatment of RF24 endothelial cells results in upregulation of PI3K/AKT signaling. A** AKT pathway activation was measured by pAKT/AKT in RF24 cells after an exogenous CD5L protein treatment. **B–D** AKT pathway activation, tube formation (**C**), and cell migration (**D**) in RF24 cells treated with CD5L protein and either LY294002 (PI3K inhibitor) or DMSO. Scale bar = 200 μm for **C** and **D**. **E** *CD36* mRNA expression in RF24 cells treated with CD5L protein. **F** AKT pathway activation in RF24 cells treated with si*CD36*. Western blots were performed from two independent technical replicates; in each repeat, the blotting, including loading control, was performed using the same sample processing controls (**A**, **B**, **F**). **G**, **H** Cell viability of RF24 cells at increasing concentrations of bevacizumab with the addition of either CD5L protein (**G**) or si*CD5L* (**H**). Data represented as mean values ± SD, determined by two-tailed Student's *t*-test except for two-way Anova Tukey's multiple comparisons for **G** and **H**. (*n* = 3 for **C**, **D**, and **E**; *n* = 4 for **G** and **H** biologically independent experiments).

been implicated as the primary cell-surface receptor for CD5L[8] and although CD36 is expressed in endothelial cells, whether CD5L played an important role in endothelial survival was not well understood[18]. We showed that silencing CD36 prevented the pro-angiogenic phenotype associated with increased CD5L expression, reinforcing the necessity of having CD5L interaction with CD36 for the development of AVA resistance.

Our analysis of an ovarian cancer cohort stratified by resistance or responsiveness to bevacizumab-based therapy demonstrated a correlation between the overexpression of CD5L in tumor endothelial cells and bevacizumab resistance. This suggests that the upregulation of CD5L by tumor endothelial cells is an important component of the adaptive resistance mechanism against bevacizumab treatment. These

findings present unique opportunities for clinical trial development aimed at preventing or reversing AVA resistance. The development of such clinical trials incorporating our RNA-aptamer or mAb against CD5L may offer additional approaches for reversing AVA resistance.

Adaptive resistance to anti-VEGF therapy is a complex mechanism programmed by tumors for continued survival. As increasing data emerge regarding the molecular pathways responsible for this phenomenon, it will be important to carefully select the critical components for the development of next-generation therapeutics and subsequent clinical trials. We recognize potential limitations, such as the lack of endothelial cells on the single-cell data and a need for additional safety testing prior to the clinical development of CD5L targeted therapy. Collectively, we have identified CD5L as an important

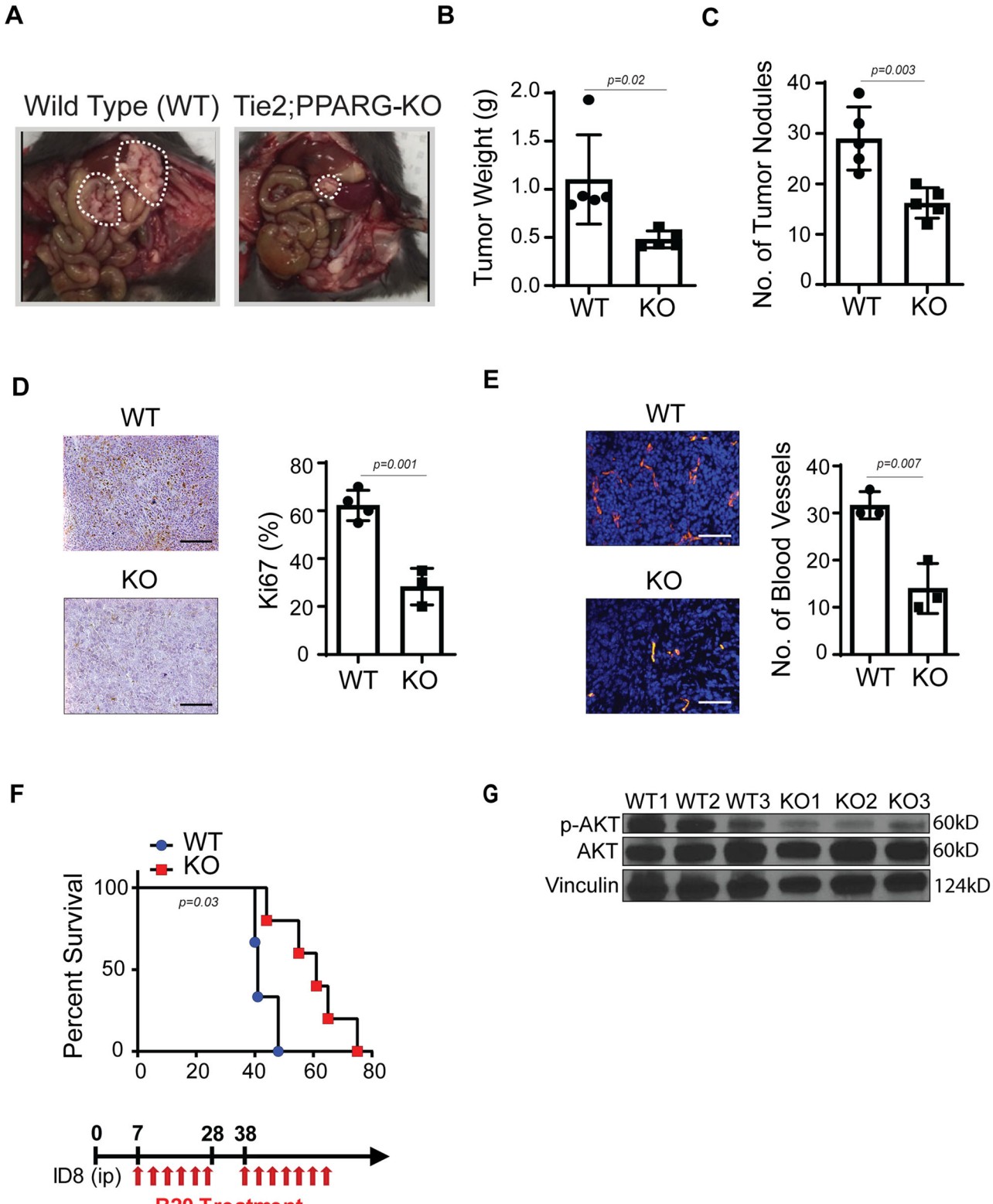

**Fig. 4 | PPARG silencing inhibits tumor growth and angiogenesis in the ID8 xenograft model. A** Photographs of representative mice from wild-type (WT) and *Tie2-cre;PPARG* KO mice. **B, C** Tumor weight (g) (**B**) and the number of tumor nodules (**C**). **D, E** Ki67 IHC (**D**) and CD31 IF (**E**) staining of tumors from WT *versus PPARG* KO mice. For statistical analysis, five randomly selected tumors per group were stained, and five random fields per tumor were scored. Scale bar = 200 μm for **D** and **E. F** Survival plot for B20, anti-VEGF antibody treatment. B20 was injected into the peritoneal cavity twice weekly at a dose of 5 mg/kg. **G** Expression of pAKT relative to AKT in tumor samples from WT versus *Tie2-cre;PPARG* KO mice (pAKT to AKT ratio determined after normalization of pAKT to AKT). Data represented as mean values ± SD, determined by two-tailed Student's *t*-test; *n* = 3–4 mice for **D, E,** and **G**; *n* = 5 mice for **B, C,** and **F**.

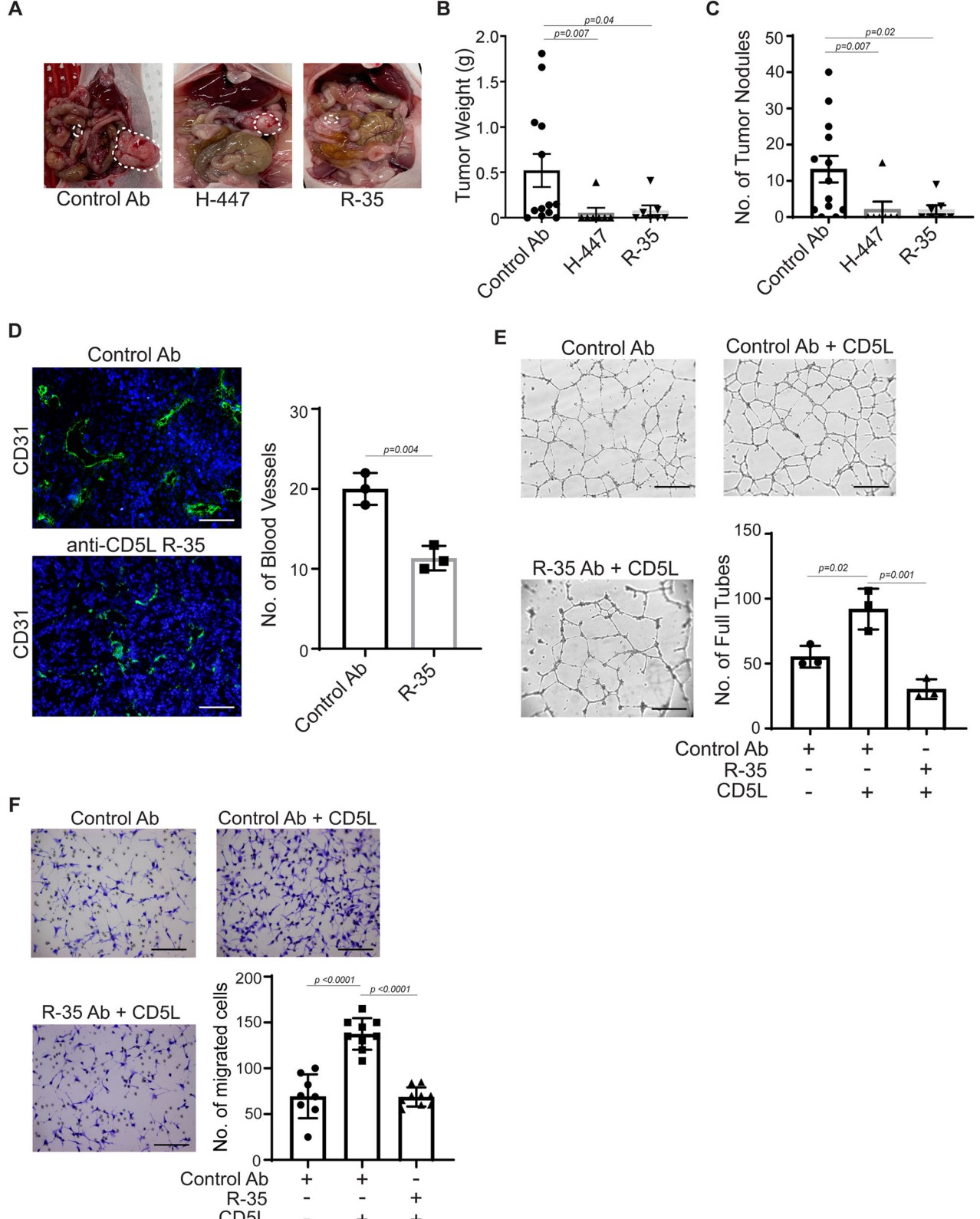

**Fig. 5 | Antibodies targeting CD5L exhibit antitumor and antiangiogenic effects. A** Photographs of representative mice of control antibody and anti-CD5L antibody (H-447 and R-35) treated groups. Mice were treated intraperitoneally with either PBS or anti-CD5L antibody (10 mg/kg) starting on Day 8 after tumor injection until Day 35. **B**, **C** Tumor weight (**B**) and the number of tumor nodules (**C**). **D** CD31 immunofluorescence staining of tumors from control versus anti-CD5L antibody-treated groups. For statistical analysis, five randomly selected tumors per group were stained, and five random fields per tumor were scored. Scale bar = 100 μm.

**E**, **F** Tube formation; scale bar = 500 μm (**E**) and cell migration; scale bar = 200 μm (**F**) of RF24 cells treated with either control antibody alone, control antibody + CD5L protein, or R-35 antibody + CD5L protein. Data represented as mean values ± SEM determined by the Mann–Whitney test for **B**, **C** ($n$ = 13 for control Ab; $n$ = 7 for H-447 and R-35 antibodies respectively), ordinary one-way ANOVA Tukey's multiple comparisons test for **E** and **F** and the two-sided Student's $t$-test used for **D**; ($n$ = 3 for **D** and **E** and 9 for **F**).

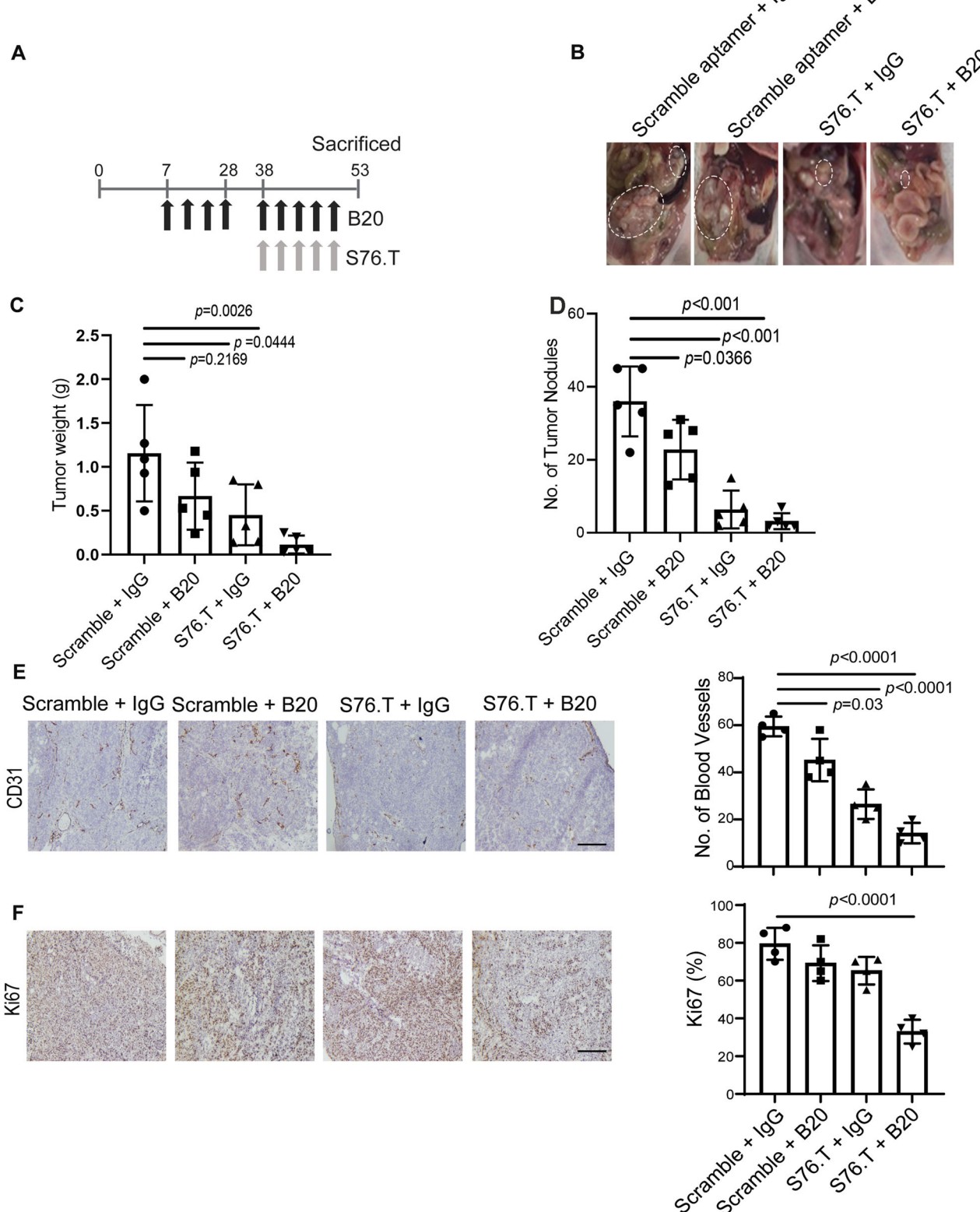

**Fig. 6 | S76.T (CD5L)-aptamer blocks resistance to anti-VEGF therapy. A** Mouse adaptive resistance tumor model. S76.T-aptamer was injected intravenously every 3 days starting on day 38, after 21 days of B20 treatment. **B** Photographs of representative mice treated with scramble aptamer, scramble aptamer + B20, CD5L aptamer (S76.T) + IgG, and S76.T + B20. **C, D** Tumor weight (g) and the number of tumor nodules. **E, F** CD31 immunohistochemical (**E**) and Ki67

immunohistochemical (**F**) staining of tumors from scramble aptamer, B20, S76.T, and combination of S76.T + B20 treated mice (scale bar = 100 μm). For statistical analysis, five randomly selected tumors per group were stained, and five random fields per tumor were scored. Data represented as mean values ± SD determined by the ordinary one-way ANOVA Tukey's multiple comparisons test; $n = 5$ for both C & D; for **D**; $n = 4$ for **E** and **F**.

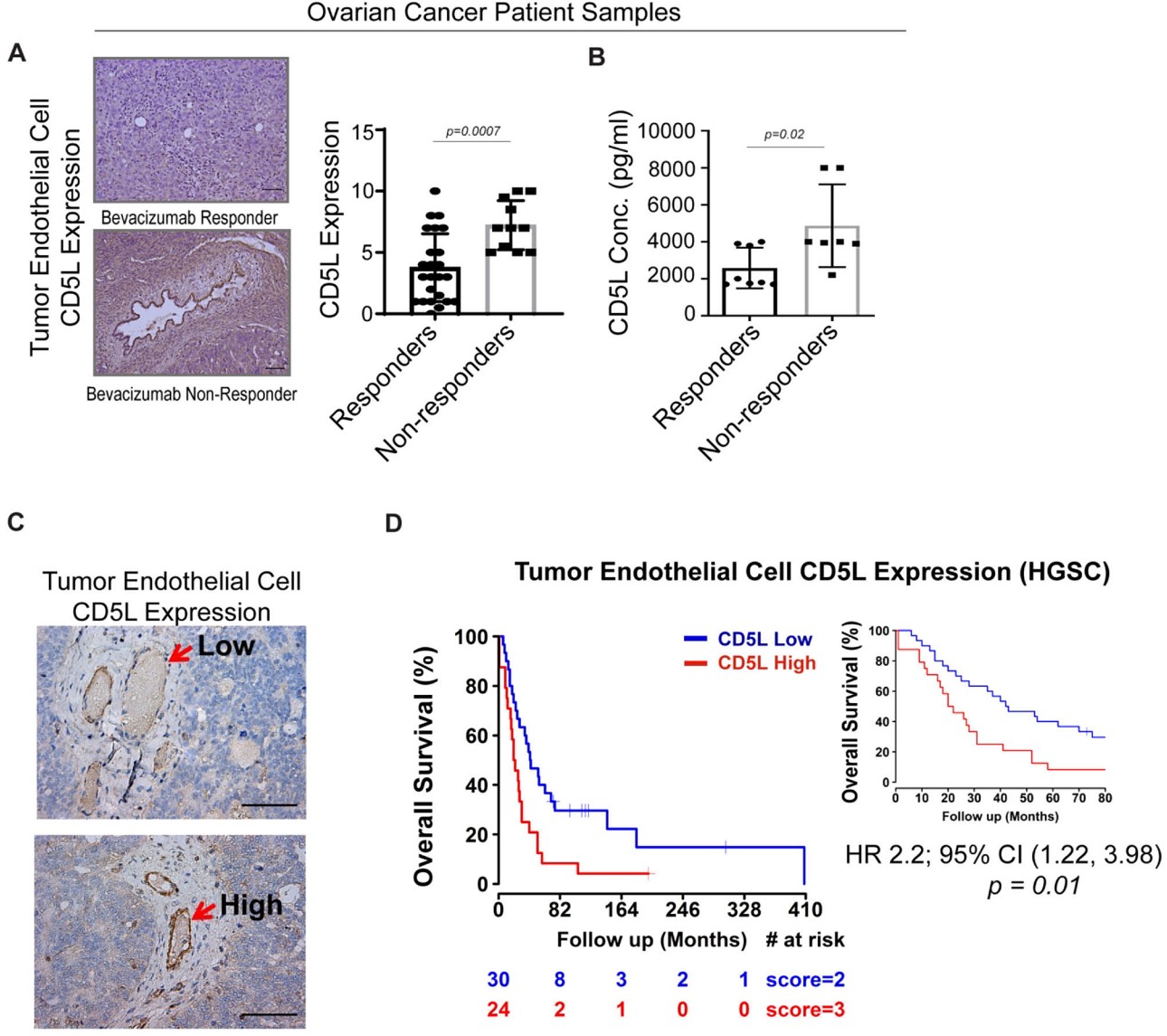

**Fig. 7 | CD5L overexpression is associated with bevacizumab resistance and worse overall survival in ovarian cancer patients. A** Representative images of CD5L expression measured by immunohistochemical (IHC) analysis in patients considered as bevacizumab responder ($n = 25$) or non-responder ($n = 11$) scale bar = 100 μm. Quantification of CD5L expression measured by IHC analysis scaled from 0 (absent) to 10 (high) in bevacizumab responders ($n = 25$) versus non-responders ($n = 11$). **B** CD5L serum protein levels in ovarian cancer patients that were classified as either responsive ($n = 8$) or non-responsive ($n = 7$) to bevacizumab. **C** Representative images of low (upper) and high (lower) CD5L protein expression in tumor endothelial cells from a human ovarian cancer patient cohort; scale bar = 100 μm. **D** Kaplan–Meier curve of overall survival in patients with high-grade serous ovarian cancer (HGSC) stratified according to CD5L protein expression level ($n = 30$ for Low and $n = 24$ for High) as measured by IHC analysis, from a patient cohort in panel (**C**). The inset on the right shows the curves from 0–80 months only. Data represented as mean values ± SD determined by a two-sided Student $t$-test was used for statistical calculations, aside from panel (**D**), which was generated with the use of the log-rank test.

protein in an adaptive response to anti-VEGF treatment. With that, strategies aimed at targeting CD5L could be of benefit to patients treated with antiangiogenic drugs.

## Methods

### Cell lines and culture

Cell lines were obtained from the MD Anderson Characterized Cell Line Core Facility, which supplies authenticated cell lines. Testing of cell lines to confirm the absence of mycoplasma, in addition to short tandem repeat DNA fingerprinting, was performed by the Cell Line Core. Human epithelial ovarian cancer cell line SKOV3ip1 and mouse ovarian cancer cell line ID8 were grown as previously described in ref. [19]. Human immortalized umbilical endothelial cells (RF24) were grown in MEM medium containing supplements (nonessential amino

acids, sodium pyruvate, MEM vitamins, and glutamine; Life Technologies, Grand Island, NY). Cell culture was performed at 37 °C in a 5% $CO_2$ incubator with 95% humidity.

For in vivo injections, cells were first washed with PBS twice, followed by trypsinization and centrifugation at under $220 \times g$ for 5 min at 4 °C. Cells were then reconstituted in serum-free Hank's balanced salt solution (Life Technologies). Only single-cell suspensions with >95% viability were used for in vivo experiments (as determined by trypan blue exclusion).

Primary endothelial cells HPAEC were grown in Endothelial Cell Growth Kit-BBE media (ATCC® PCS-100-040) with a full set of growth factors. Venous endothelial cells (HUVEC) were grown in EBM-2 basal medium and EGM-2 singleQuots supplements. Human immortalized umbilical endothelial cells were grown in MEM medium containing

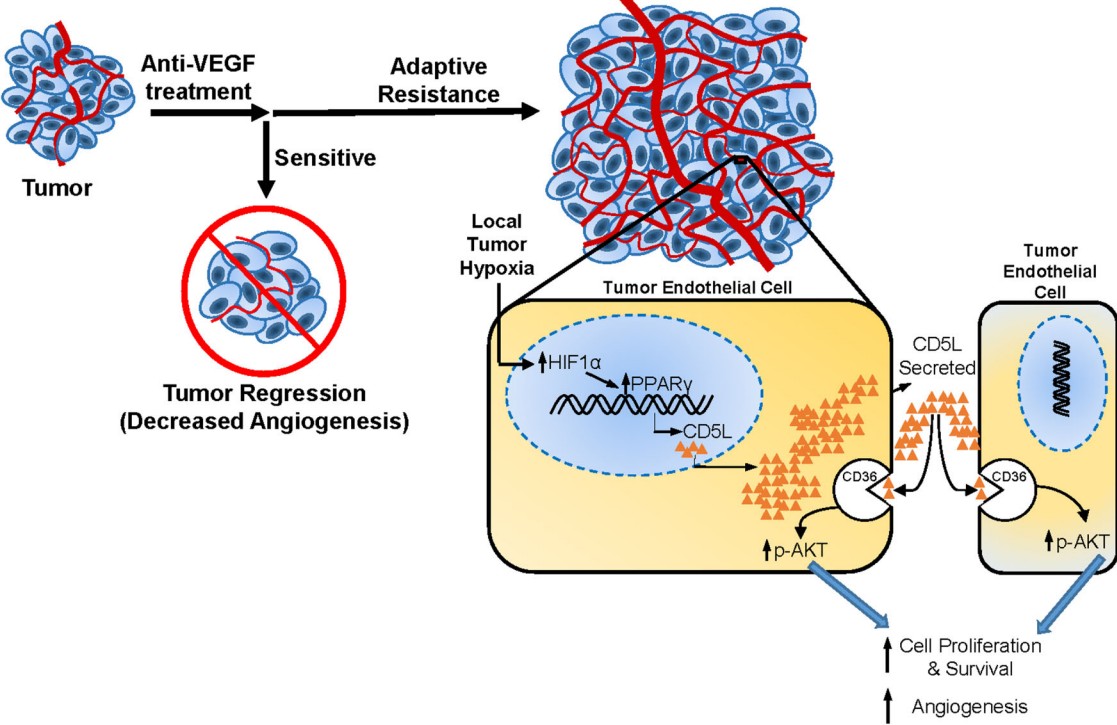

**Fig. 8 | Mechanism of CD5L-induced AVA resistance.** Anti-VEGF treatment may initially cause tumor regression via decreased angiogenesis (tumor with low vessel density); however, adaptive resistance frequently emerges over time, leading to tumor growth and increased angiogenesis (larger tumor with high vessel density). Inset demonstrates tumor endothelial cells showing that local tumor hypoxia leads to increased CD5L secretion by overexpression of transcription factor PPARG. Secreted CD5L binds to the CD36 receptor, causing activation of the AKT pathway and ultimately leading to increased cell proliferation and angiogenesis.

supplements (nonessential amino acids, sodium pyruvate, MEM vitamins, and glutamine; Life Technologies). Mouse ovarian tumor ID8 cells were grown in DMEM supplemented with 5% serum and ITS (mixture of recombinant human insulin, human transferrin, and sodium selenite). Cell culture was performed at 37 °C in a 5% $CO_2$ incubator with 95% humidity.

### Quantitative real-time reverse-transcriptase polymerase chain reaction validation

Quantitative real-time reverse-transcriptase polymerase chain reaction (qRT-PCR) was conducted by using 50 ng of total RNA isolated from cells with the use of the RNeasy mini kit (Qiagen, Germantown, MD). Complementary DNA (cDNA) was synthesized from 0.5–1.0 μg of total RNA with the use of a Verso cDNA kit (Thermo Scientific, Waltham, MA). qPCR analysis was performed in independent triplicate by using the reported primers (Table S3) and the SYBR Green ER qPCR Super-Mix Universal (Invitrogen, Carlsbad, CA) with the Applied Biosystems ABI 7500 (Applied Biosystems, Waltham, MA). Quantification was performed by using the $2^{-\Delta\Delta CT}$ method normalizing to control for percent fold changes[20].

### Cell proliferation assay

Cell proliferation experiments were performed by using the Click-iT EdU Assay Kit (Invitrogen). Cells were seeded into 6-well plates and cultured in a phenol-red free Opti-MEM for 48 h. Cells were then harvested for assessment of proliferation after CD5L protein or si*CD5L* treatment.

### Enzyme-linked immunosorbent assay

CD5L expression levels were determined with the use of a CD5L ELISA Kit (MyBioSource, San Diego, CA) according to the manufacturer's recommendations for both cell culture media as well as human serum samples.

### Lentivirus-mediated CD5L overexpression

The pLenti-C-mGFP-human CD5L vector (RC206528L2) was purchased from Origene (Rockville, MD). HEK293T cells were co-transfected with the pLenti-C-mGFP-human CD5L vector and packaging plasmids. After 48 h, the supernatant containing infectious viral particles was collected, filtered with 0.45-μm filters, and stored in aliquots at −80 °C. To generate cells that ectopically overexpressed CD5L, RF24 cells were incubated for 24 h with viral particles. Cells were then washed with PBS and further incubated in a culture medium. After 48 h, GFP-positive cells were sorted with the use of a FACS Aria II sorter (Beckton, Dickinson and Company, Franklin Lakes, NJ).

### Induction of bevacizumab resistance in RF24 cells

Bevacizumab-resistant cells were derived from the original parental RF24 cell line by continuous exposure to bevacizumab. RF24 cells were treated with bevacizumab (1 mg/ml, IC50) for 72 h. This medium was then removed, and the cells were allowed to recover for 7 days. Cells were continuously maintained in the presence of bevacizumab at IC50 concentrations.

### Drug sensitivity assay (MTT)

Cells ($5 \times 10^3$) were seeded in 96-well plates and allowed to adhere overnight at 37 °C. Briefly, after treatment of cells with bevacizumab for 72 h, MTT reagent [(3-(4,5-dimethylthiazol-2-yl)−2,5-diphenyl tetrazolium bromide)] was added to each well and incubated for 2 h at 37 °C. Dimethyl sulfoxide (DMSO) was added to each well and mixed for 5 min on an orbital shaker. Absorbance was recorded at 450 nm, and sensitivity to bevacizumab was calculated based on cell viability measurements at 72 h.

### Cell migration assay

We examined the migration of RF24 cells in the presence or absence of h*CD5L* siRNA, CD5L aptamer, or anti-CD5L antibody with the use of

transwell inserts (Corning, Lowell, MA) coated with 0.1% gelatin. After transfection of 48 h with h*CD5L* siRNAs or after treatment with either CD5L aptamer or anti-CD5L antibody (40 μg/ml), RF24 cells (1.0 × 10[5]) in MEM serum-free medium with CD5L protein (200 ng/ml) were seeded into the upper chamber of the transwell. The insert was then placed in a 24-well plate containing MEM medium with 15% serum in the lower chamber as a chemoattractant. After cells were allowed to migrate for 6 h in a humidified chamber, those that had migrated were stained with a HEMA3 staining kit (Life Sciences, Durham, NC) and counted by light microscopy in five random fields (×200 original magnification) per sample. Experiments were done in duplicate and repeated three times.

## Tube formation assay
Matrigel (12.5 mg/ml) was thawed at 4 °C, and either 50 μL was quickly added to each well of a 96-well plate or 10 μL to each well of a μ-Slide 15-Well ibidi plate and then allowed to solidify for 30 min at 37 °C. RF24 cells were subjected to the various treatments described below and added to each well (5000–6000 cells/well) and then incubated for 6 h at 37 °C. Treatment 1: RF24 cells were pretreated with siControl or si*CD5L* for 48 h and then added to each well. Treatment 2: RF24 cells were treated with 200 ng of CD5L protein for 2 h, followed by LY294002 (50 μM) for 6 h, and then added to each well. Treatment 3: RF24 cells were simultaneously treated with both CD5L protein and CD5L aptamer and then directly added to each well. Treatment 4: RF24 cells were simultaneously treated with both CD5L protein and anti-CD5L antibody (40 μg/ml) and then directly added to each well. Experiments were performed in triplicate and repeated at least twice. With the use of an Olympus IX81 inverted microscope, five images per well were taken at ×100 magnification. The number of nodes (defined as at least three cells that formed a single point) and tubes (defined as a non-segmented circle formed from endothelial cells) per image was quantified. The highest and lowest values were removed from each group to account for cell clumping.

## Spheroid formation using the hanging-drop method
GFP-labeled HUVEC cells were grown in EBM-2 basal medium and EGM-2 singleQuots supplements. Lids of 10 cm cell culture dish were seeded with 20 μl growth medium containing 400 cells and then inverting culture dish lid. For hydration of the drops, 10 ml PBS were added to the bottom of the cell culture dish. Cells were incubated at 37 °C and 5% $CO_2$ for overnight to allow aggregation. The HUVEC spheroids were collected in a 1.5 ml tube and suspended in 1 ml of fibrin solution (2 mg/ml), followed by dispensing in 24-well plates containing 500 μl of thrombin (1U/ml). The fibrin solution with spheroids was mixed gently with thrombin and incubated for 30 min at 37 °C. EBM-2 basal medium with or without CD5L protein (400 ng/ml) was applied in each well on the top of the gel. Spheroids were also treated with a control antibody or anti-CD5L antibody (R-35, 40 μg/ml) for 24 h, and sprouting was determined by quantifying the area surrounding the sprout outgrowth with the Olympus CKX41 imaging system.

## CD5L expression in proliferating and resting conditions
Arterial (HPAEC) and venous (HUVEC) primary endothelial cells were serum starved for 72 h in BBE serum-free media with supplements. *CD5L* and *p27* expression was checked in proliferating (PEC) and resting (QEC) conditions using qRT-PCR.

## siRNA constructs and delivery
siRNAs were obtained from Sigma-Aldrich (The Woodlands, TX). Control siRNA consisted of a non-silencing siRNA that did not share sequence homology with any known human mRNA based on a BLAST search. In vitro transient transfection was performed according to the manufacture's guideline (https://www.sigmaaldrich.com/deepweb/

assets/sigmaaldrich/product/documents/400/389/s1452bul.pdf). In brief, siRNA (2 μg) was incubated with 6 μL of Lipofectamine 2000 transfection reagent (Invitrogen) for 20 min at room temperature and then added to cells cultured in 10 cm plates at 60% confluence.

## Reverse phase protein array (RPPA) and western blot analysis
RF24 cells were cultured in the presence or absence of human recombinant CD5L protein (Sino Biological, Beijing, China). The corresponding cell lysate was then submitted for RPPA analysis performed at the Functional Proteomics RPPA Core Facility at the MD Anderson Cancer Center. Western blot analysis was performed according to the manufacturer guideline (https://www.sigmaaldrich.com/US/en/technical-documents/protocol/protein-biology/western-blotting/western-blotting). Cell lysate of RF24 cells was collected after treatment with human recombinant CD5L protein, and activation of AKT signaling was checked by Western blotting by using anti-human CD5L, pAKT, AKT, and PPARG antibodies, followed by appropriate secondary antibodies conjugated with horseradish peroxidase. Experiments were done in duplicate and repeated at least twice. Uncropped and unprocessed scans of the Western blots are provided in the Source Data file.

## Promoter analysis and chromatin immunoprecipitation (ChIP) assay
RF24 cells were cultured in a hypoxic condition for 16 h. After hypoxic culture, ChIP assays were performed by using an EZ ChIP™ kit (Millipore, Temecula, CA) according to the manufacturer's instructions. In brief, cross-linked cells were collected, lysed, sonicated, and subsequently subjected to immunoprecipitation with PPARG (Cell Signaling) antibody or IgG control. Immunocomplexes were collected with protein G agarose beads and eluted. Cross-links were reversed by incubating at 65 °C. DNA was then extracted and purified for subsequent PCR amplification with the use of gene-specific primers. Negative control forward: GCAGTGAGAAAGCAGGTTTG, negative control reverse: CATAATCCAGGGTCATGGTG. PPARG binding region forward: TCCTTTCCTCTGGAACATGC, PPARG binding region reverse: TAAGAGGAGGGACAAAGACAGG.

## Microarray analysis
We extracted total RNA from B20-sensitive and B20-resistant tumor endothelial cells with the use of an RNeasy mini kit (Qiagen). A Nano-Drop spectrophotometer (Thermo Fisher Scientific) was used for the assessment of both RNA quality and quantity. Total RNA (700 ng) was labeled and hybridized to Bead Chips according to the manufacturer's protocols (Illumina, San Diego, CA). We scanned Bead Chips with an Illumina BeadArray Reader, and all array normalization, filtering, and statistical analysis was performed by using BRB-ArrayTools (National Cancer Institute). The microarray data were submitted to GEO under the accession number GSE180687.

In brief, gene lists were generated according to the following parameters: spots were excluded if the spot intensity was below a minimum value of 100 and if the detection call contained the value "A". Genes were excluded if the percentile of the log-ratio variation was <50%, the percentage of data missing or filtered out exceeded 50%, or the percentage of absent (i.e., detection call = A) data exceeded 50%. The number of genes used for random variance estimation was 16,645, and the number of genes that passed the filtering criteria was 14,213. A two-sample Student $t$-test (with random variance model) was then used to determine statistical significance with a $p$ value set to <0.05.

## Toxicity assessment of CD5L blocking antibody on normal mice and endothelial cells
Wild type C57BL/6 mice were treated either with control IgG or R-35 for two weeks (once a week). Blood was collected and CBC (complete blood count) was performed. H&E sections of lung, liver, kidney, and spleen tissue were examined by two in-house pathologists. Blood

 

vessels in organs were checked by staining the blood vessels with mouse-CD31 antibody. CD5L expression in mouse organs was checked by staining the tissue with CD5L and CD31 antibodies.

## CD5L expression in different tumor cell types

The sample collection for single-cell analysis was approved by the Institutional Review Board of The University of Texas MD Anderson Cancer Center. The analysis includes chemo-naïve patients diagnosed with advanced high-grade serous ovarian carcinoma and tissue from the primary and metastatic stites was used. The fresh tissue was obtained from the MD Anderson Tumor bank and sample processing was done right after tissue collection from the patient. About 2 g of tissue were minced and dissociated using Accumax™. After dissociation, cells were washed with complete media to stop the reaction, centrifuged, and resuspended in freeze media (90% FBS/ 10% DMSO) and frozen at −80 °C. For the single-cell protocol, cells were thawed and stained with LIVE/DEAD Aqua™ and CD45 and sorted using BD FACSAria™. After sorting, cells were processed following the 10x genomics Chromium Single Cell 3′v3 chemistry protocol. The single-cell data were submitted to GEO under the accession number GSE181955. The data was analyzed using CellRanger software version 3.1.0, and a gene expression matrix based on universal molecular identifier (UMI) counts was generated. We further employed Seurat version 3.1.4 for unbiased cell clustering analysis. An example of the gating strategy is illustrated in Fig. S19A–D. Briefly, we first filtered the matrix based on minimum/maximum cut-offs for genes/cell, cells/gene, and optional parameters such as mitochondrial gene UMI count as a percentage of the total and normalized the data. Then, the variable genes were identified and the data was scaled as previously described (https://hbctraining.github.io/scRNA-seq_online/lessons/06_SC_SCT_normalization.html). Next, we identified principal components; and finally identified cell "neighbors" and cell clusters.

## Receptor-mediated Anti-CD5L antibody (R-35) specificity using endothelial-specific CD36 knockout mouse model (CD36$^{flox/flox}$Tie2$^{Cre}$ GEM)

All studies were approved and supervised by the MD Anderson Institutional Animal Care and Use Committee (IACUC). Luciferase-labeled ID8-Luc mouse ovarian cancer cells ($1.0 × 10^6$ cells) were inoculated intraperitoneally into age-matched (4 to 6 weeks old), female C57BL/6 (Taconic Biosciences), and CD36$^{flox/flox}$Tie2$^{Cre}$ mice. After the initial establishment of tumors as measured by IVIS imaging ($1 × 10^4$ photons/second/cm$^2$/sr), (-day 8), mice were treated with either anti-CD5L antibody (R-35) to groups (1) WT ($n = 10$), (2) CD36$^{flox/flox}$Tie2$^{Cre}$($n = 8$); or control IgG to group (3) CD36$^{flox/flox}$Tie2$^{Cre}$($n = 8$) for 4 weeks. Tumor progression was monitored by weekly bioluminescence. Animals were sacrificed when tumor burden exceeded the established guidelines by IACUC (~35 days post-inoculation). Tumor weight, the number of tumor nodules, volume of ascites fluid, and body weight of each mouse were recorded at the time of necropsy. Data were expressed as mean ± SD, determined by a two-tailed, nonparametric $t$-test. Representative gross images from ~6-week-old C57BL/6, CD36$^{flox/flox}$Tie2$^{Cre}$ mice that received R-35 or control IgG antibody treatment were shown in the manuscript. Gross images for bioluminescence in C57BL/6 or CD36$^{flox/flox}$Tie2$^{Cre}$ mice were recorded and the quantification of bioluminescence imaging for the three groups of mice were performed (luminescence counts × $10^4$ photons/second/cm$^2$/sr).

## Fatty acid uptake in CD36 KO endothelial cells

To assess fatty acid uptake, $6 × 10^3$ RF24 or siRNA-CD36 transfected cells were seeded in 100 μL full medium containing 10% FBS in a 96-well plate. The CD36 expression was silenced in RF24 cells using CD36 siRNA (Sigma; SASI_HS01_0057_5562) with X-tremeGENE HP DNA Transfection Reagent (Promega Fugene HD Transfection Reagent, REF E2312). The knockdown efficiency of siRNAs against CD36 were

previously validated via western blotting using an anti-CD36 antibody (Abcam Inc, REF ab252922).

After 48 h at 37 °C, 5% $CO_2$, the cells were treated with either 40 μg/mL of R-35 or 200 ng/ml of rCD5L for 6 h. The cells were washed twice with PBS and serum-deprived for 1 h before adding 100 μL TF2-C12 Fatty Acid Stock Solution (Sigma-Aldrich, MAK156). The fluorescence signal was measured at Ex/Em = 485/515 nm after incubating the cells for 30 and 60 min at 37 °C. Each condition was repeated six times ($n = 6$), and three empty wells, including a full amount of TF2-C12, were included as normalization controls.

## Experimental anti-CD5L antibody generation

Monoclonal antibodies were screened and selected from two antibody sources. One source was from CD5L immunized rabbit single B cells. Briefly, two New Zealand white rabbits were immunized with 0.5 mg of recombinantly expressed human CD5L protein (Sino Biological). After the initial immunization, animals were given three boosters in a 3-week interval until serum titers reached $10^6$. Single B cells were isolated and screened for CD5L-binding antibodies by using ELISA for initial positive hits. A human-naïve scFv phage display library ($10^{10}$ diversity) was also used as another source for the selection of monoclonal antibodies against CD5L. The solid phase panning method was used for the selection of CD5L binders by ELISA[21]. Antibody variable coding regions in the positive hits were cloned and sequenced according to a method reported previously[22]. Full-length antibody heavy and light chain constructs were made with human IgG constant region sequences in fusion with cloned variable sequences for expression of monoclonal antibodies in human embryonic kidney freestyle 293 (HEK293F) cells (Life science Technologies). Antibodies were purified with the use of A/G affinity resin to purity >95% as reported previously[23].

## Orthotopic in vivo model of ovarian cancer

Female athymic nude mice (NCr-nu) and C57BL/6 wild-type mice were purchased from Taconic Biosciences (Rensselaer, NY). All mouse studies were approved by the Institutional Animal Care and Use Committee. Mice were cared for in accordance with guidelines set forth by the American Association for Accreditation of Laboratory Animal Care and the US Public Health Service Policy on Humane Care and Use of Laboratory Animals. For the SKOV3ip1 model, adaptive resistance to anti-VEGF antibody therapy was established by injecting luciferase-labeled cells, as described previously in ref. [24]. After 2–3 weeks, each mouse received initial treatments with a control IgG antibody or B20, a murine monoclonal VEGF-A antibody (5 mg/kg, injected intraperitoneally twice weekly), and tumor growth was monitored by IVIS imaging (Xenogen, Alameda, CA). After 3–6 weeks, the mice were placed in B20-sensitive and B20-resistant groups according to their responses to B20 treatment on tumor growth (bioluminescence signal).

For the CD5L aptamer therapy experiments, aptamer (administered intravenously) and B20 (administered intraperitoneally) were given once weekly at a dose of 5 mg/kg body weight. For the anti-CD5L antibody experiments, SKOV3ip1 cells ($1 × 10^6$) were injected into the peritoneal cavity. Control or experimental antibody was injected intraperitoneally once weekly at a dose of 10 mg/kg body weight.

Mice were euthanized with $CO_2$ when they became moribund in any group due to high tumor burden. After the mice were euthanized, we recorded their tumor weight and the number and distribution of tumor nodules. Individuals who performed the necropsies were blinded to the treatment group assignments. Tissue specimens were either fixed by using 10% buffered formalin, frozen in OCT (Miles, Inc., Elkhart, IN), or snap-frozen in liquid nitrogen.

## Endothelial cell isolation

Tumor tissues harvested from B20-sensitive or B20-resistant groups were minced and digested with elastase, collagenase A, and DNase 1 at

37 °C for 90 min to obtain a single-cell suspension. Platelets and RBCs were removed by performing Percoll separation, and tumor cells were sorted by FACS by using CD326 antibody (Miltenyi Biotec, Auburn, CA) as a positive selection marker for tumor cells. Endothelial cells were sorted by FACS using CD31 antibody (Miltenyi Biotec) as a positive selection marker for endothelial cells, according to the manufacture guidance (https://www.miltenyibiotec.com/US-en/applications/all-protocols/isolation-and-cultivation-of-endothelial-cells-from-adult-mouse-brain.html#gref).

### Immunohistochemical and immunofluorescence staining of xenografts

Immunohistochemical analyses for cell proliferation (Ki67, 1:200, Zymed, San Francisco, CA) and immunofluorescence analyses for mean vessel density (CD31, 1:800, Pharmingen, San Diego, CA) were performed as described previously in ref. [25]. For statistical analyses, sections from five randomly selected tumors per group were stained, and five random fields per tumor were scored. Pictures were taken at ×200 or ×100 magnification. To quantify mean vessel density in the mouse tumor samples, the number of blood vessels that stained positive for CD31 was recorded in five random $0.159\text{-}mm^2$ fields at ×200 magnification. To quantify Ki67 expression, the number of positive cells was counted in five random $0.159\text{-}mm^2$ fields at ×100 magnification[25]. All staining was quantified by two investigators in a blinded fashion.

For human ovarian cancer specimens, CD5L expression was determined according to standard immunohistochemical procedures as previously described in ref. [26]. Scoring of CD5L expression within tumor endothelial cells of bevacizumab responder versus non-responder patients was performed according to the following rubric with the use of ImageJ software (National Institutes of Health, Bethesda, MD): 0, non-stained vessel; 0.5–3.0, weakly stained vessel; 3.5–7.0, moderately stained vessel; 7.5–10.0, strongly stained vessel. CD5L expression in tumor endothelial cells from our TMA ovarian cancer cohort was scored as either 1 (negative), 2 (low), or 3 (high).

### Generation of CRISPR/Cas9-PPARG and CD5L knockout (KO)-RF24 endothelial cells

*CD5L* - KN2.0, Human gene knockout kit via CRISPR, non-homology mediated construct KN406528 Origene, and *PPARG* - KN2.0, Human gene knockout kit via CRISPR, non-homology mediated KN401538 Origene) were used to generate the stable knockout of *CD5L* or *PPARG* in RF24 endothelial cells. The transfections of two gDNA or scramble controls with donor vectors in each CRISPR/Cas9 kit were delivered by X-tremeGENE HP DNA Transfection Reagent and Opti-MEM I (Life Technologies). 48 h post-transfection, cells were passaged based on the manufacturer's recommendation and selected with 0.33 μg/ml puromycin. The efficacy of CRISPR/Cas9 knockout of *CD5L* or *PPARG* was validated by western blots. *CD5L* KO or *PPARG* KO endothelial cells were treated with CD5L recombinant protein and checked tube formation.

### Tie2-cre;PPARG knockout mice

*PPARG*$^{fl/fl}$*Tie2-cre*$^{+/-}$ female and male mice were generously provided by Dr. Yihong Wan (University of Texas Southwestern). To selectively delete PPARG in endothelial cells, female and male mice were crossed to obtain *PPARG* conditional knock-out mice. Genomic DNA was isolated from the tail biopsies of each mice. *Tie2-cre* transgene and floxed *PPARG* alleles were distinguished by PCR by using primers (Supplementary Table 3). For tumor cell injection, ID8 cells ($1 \times 10^6$) were injected intraperitoneally in *Tie2-cre*; *PPARG* KO mice. Mice were euthanized when they became moribund in any group due to high tumor burden and/or ascites.

For the survival experiment, percent survival was determined based on animals becoming moribund individually. After mice were

euthanized with $CO_2$, their tumor weight, number, and distribution of tumor nodules were recorded. Individuals who performed the necropsies were blinded to the treatment group assignments. Tissue specimens were collected as mentioned above. For B20 therapy experiments, B20 was given twice weekly at a dose of 5 mg/kg body weight after 7 days from ID8 tumor injection. Depending on the experimental design, B20 treatment continued either until the planned endpoint was reached or until mice became moribund or deceased.

### Human ovarian cancer samples

Tumor tissue and serum were obtained from 25 ovarian cancer patients classified as responsive to bevacizumab (partial or complete response) and 11 patients classified as non-responsive to bevacizumab (stable or progressive disease). In addition, tumors from 116 patients with serous ovarian cancer diagnosed between January 1, 1985, and March 31, 2004, were evaluated with the use of a tissue microarray (TMA) obtained from Wayne State University. Each tumor was represented as two cores within the TMA, and the diagnosis was confirmed by a board-certified pathologist. Clinical outcome data were available and used to generate Kaplan–Meier curves (R version 3.4.1) based on stratification of specific histology and CD5L tumor endothelial cell expression scores. Approval for both sets of human samples was provided by the Institutional Review Board (IRB).

### Aptamer

S76.T and the unrelated scrambled, used as a negative control, are custom synthesized 2′-Fluoro Pyrimidines (2′-F Py) modified RNAs (Sigma-Aldrich, St. Louis, MO).

S76.T: 5′ AGGUUGCAGCGUUCGACAGGAGGCUCACAACAG 3′
Scrambled of unrelated aptamer:
5′ UUCGUACCGGGUAGGUUGGCUUGCACAUAGAACGUGUCA 3′

Before each treatment, S76.T and the control aptamers were subjected to a denaturation–renaturation step using the following protocol: 85 °C for 5 min, cool down on the ice for 2 min, and warm up to 37 °C.

### Tandem protein-systematic evolution of ligands by exponential enrichment

The tandem variant of the protein–Systematic Evolution of Ligands by Exponential Enrichment (SELEX) procedure consists of two independent in vitro selections performed in tandem. The counter-selection/selection rounds were repeated for eight cycles for the first round of selection followed by two cycles for the second selection.

The starting DNA library of the first selection contains $10^{15}$ different random sequences with 2′-Fluoro-Pyrimidine (2′-F Py) modification (TriLink Biotechnologies, San Diego, CA). The DNA library consisted of a random internal region of 40 base pairs flanked by two constant regions at the 5′ and 3′ ends for the amplification reaction. The DNA library was prepared as described in refs. [27,28]. However, the starting RNA pool of the second cycle of selection was the 2′-F Py selected sequence S5.

His-tagged CD5L (Sino Biological Inc., Beijing, China) was used as the target for the selection step and His-tagged VEGF-A (Abnova, Taiwan), His-tagged CD19 (Life Technologies, Carlsbad, CA), and 6-His Peptide tagged (BioLegend, San Diego, CA) for the counter-selection step. To enhance the specificity of aptamers for the target protein, the RNA pool was incubated with His-tagged counter-selection protein for 30 min (counter-selection step). The unbound sequences were incubated with the target protein CD5L (selection step) at room temperature. The RNA–protein complexes were recovered by incubation with $Ni^{2+}$ NTA Magnetic Agarose Beads (Qiagen) for 30 min. The bound sequences were isolated by total RNA extraction by using TRIzol Reagent (Life Technologies), reverse transcribed with M-MuLV Reverse Transcriptase (Roche, Indianapolis, IN), and amplified by PCR (1 min at

93 °C, 1 min at 53 °C, and 1 min at 72 °C) under high MgCl$_2$ and dNTP concentrations to introduce random mutations into the sequences.

The DNA template was in vitro transcribed to RNA. Before each round of SELEX, 2′-F Py RNA pool was subjected to a denaturation–renaturation as described above. The binding buffer used consisted of 10 mM Tris-HCl 7.5, 150 mM NaCl, 1 mM MgCl$_2$, 0.1% Triton. To enhance the stringency, progressively, the number of washings was increased, and the RNA/protein molar ratio was reduced. The final pool was cloned using TOPO-TA (TOPO Cloning, Life Technologies, Carlsbad, CA) and transformed into *Escherichia coli*. The clones were isolated and sequenced (www.eurofinsdna.com). ClustalW2 software was used for sequence analysis and alignments, and the TreeVieX program was used to visualize the phylogenetic tree. RNA structure software (version 5.1) was used to predict the secondary structures.

### Microscale thermophoresis (MST)

Four sequences from the first (S5, S72, S47, and S63) and the second selection (S11, S23, and S76) were tested to determine the binding affinity by MTS. A serial dilution of S5, S47, S63, and S72 was prepared in a specific buffer composed of PBS pH 7.4, 1 mM MgCl$_2$, and 0.05% Tween 20. The highest concentration used was 18,000 nM and the lowest was 0.54 nM. About 4 μl of each dilution step were mixed with 4 μl of the labeled molecule CD5L purified recombinant protein (Sino Biological Inc., Beijing, China) or VEGF scrambled protein used as the negative control (R&D System) (constant concentration of 12.5 nM).

The four different aptamers from the first selection were tested for binding for CD5L and VEGF protein (Table S1). Regarding the three sequences of the second selection (Table S2), S11 and S23 sequences were incubated with a constant concentration (1 nM) of labeled CD5L purified recombinant protein (Sino Biological Inc., Beijing, China) or labeled VEGF scrambled protein used as negative control (R&D System). S11 and S23 were diluted 1:1 (from 15,000 to 0.05 nM) in binding buffer (pH 7.4, 1 mM MgCl$_2$, 0.1% Tween 20). The S76 sequence was diluted 1:1 in Tris-buffered saline (TBS) from 1000 nM to 0.03 nM. S76.T was labeled with Cy5, and the concentration was kept constant at 2 nM. CD5L was diluted from 2000 nM to 0.06 nM in 50 mM Tris-HCl pH 7.8, 150 mM NaCl, 0.05% Tween 20. 5 μL of each dilution step was mixed with 5 μL of the labeled CD5L or VEGF. The final reaction mixture contained a respective amount of aptamer sequences and a constant 1 nM labeled CD5L or VEGF.

For Cy5 labeled S76.T, a serial dilution of CD5L was prepared in the buffer composed of 50 mM Tris-HCl pH 7.8, 150 mM NaCl, 10 mM MgCl2, 0.05%. Tween 20. The highest concentration of CD5L was 2000 nM and the lowest was 0.06 nM. About 4 μL of each dilution step were mixed with 4 μL of the Cy5 labeled S76 truncated (4 nM stock). The final reaction mixture, which was filled in standard capillaries, contained a respective amount of CD5L (max. conc. 1000 nM, min conc. 0.03 nM) and constant 2 nM labeled S76 truncated MST measurement was obtained using Monolith NT.115 (NanoTemper Technologies GmbH, Munich, Germany) with standard capillaries. Kd was determined as described previously in ref. [29].

### Statistical analyses

Kaplan–Meier survival curves were generated and compared with the use of log-rank statistics to assess the effect of tumor vascular CD5L expression on human overall survival and to determine survival in the *Tie2-cre;PPARG* KO mouse model. For the animal experiments in Figs. 4, 6, ten mice were assigned per treatment group. This sample size gave 80% power to detect a 50% reduction in tumor weight with a 95% confidence interval. Since the in vivo experiment in Fig. 5 was designed to screen antibody candidates, only seven mice were assigned per treatment group. Tumor weights and the number of tumor nodules for each group were compared by using either the

Student *t*-test (for comparisons of two groups) if the distribution was normal or Mann–Whitney if the distribution was not normal. A *P* value of less than 0.05 were deemed statistically significant. All statistical tests were two-sided and were performed by using either SPSS version 12 for Windows statistical software (SPSS, Inc., Chicago, IL) or Graph-Pad Prism 7 for Windows (GraphPad Software, La Jolla, CA). Kaplan–Meier curves were generated by using R version 3.4.1 (R Foundation for Statistical Computing, Vienna, Austria).

### Reporting summary

Further information on research design is available in the Nature Portfolio Reporting Summary linked to this article.

## Data availability

The microarray data and the single-cell data generated in this study have been deposited in the GEO under the accession numbers GSE180687 and GSE181955. All remaining data associated with this study are available within the article. Source data are provided with this paper.

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

## Acknowledgements

Portions of this work were supported by the Department of Defense Ovarian Cancer Research Program (W81XWH-20-1-0335 Y.W); the National Comprehensive Cancer Network (Y.W.); the Marsha Rivkin Center for Ovarian Cancer Research (Y.W.). C.J.L. was supported by the T32 CA101642 from the NIH. We thank the Italian Ministry of Economy and Finance to the CNR for the Project FaReBio di Qualità (VdF). S. P. was supported by the Ovarian Cancer Research Fund Alliance (OCRFA). M.H. was supported by CPRIT Single Cell Genomics Center Grant (RP180684). A.C.-R. is supported by funds from Deutsche Forschungsgemeinschaft (CH 1733/1-2). C.-A.M. was supported by the National Institute of Health-National Eye Institute (EY024376); the National Eye Institute Vision Core Grant P30EY028102. The RPPA Core is supported by NCI Grant # CA16672 and Dr. Yiling Lu's NIH R50 Grant # R50CA221675: Functional Proteomics by Reverse Phase Protein Array in Cancer. We are indebted to Dr. Thomas Shubert of 2Bind, who determined the binding affinity of S76 by microscale thermophoresis analysis. We also thank the MD Anderson Flow Cytometry and Cellular Imaging Core Facility (funded by NCI Cancer Center Support Grant P30 CA16672) and MD Anderson's Research Medical Library (Tamara K. Locke) for editing this manuscript. This work was also supported, in part, by the NIH (P50 CA217685, and R-35 CA209904), The University of Texas MD Anderson Ovarian Cancer Moonshot, Ovarian Cancer Research Fund Alliance, the Dunwoody Fund, the Blanton-Davis Ovarian Cancer Research Program, American Cancer Society Research Professor Award, Judy's Mission, and the Frank McGraw Memorial Chair in Cancer Research. This work was also partly supported by the Cancer Prevention and Research Institute of Texas (RP150551, and RP190561), the Welch Foundation (AU-0042-20030616) to Z. A.

## Author contributions

C.J.L.: Formal analysis, validation, methodology, and wrote the original draft; P.A.: Formal analysis, validation, investigation, visualization, methodology, and writing; K.N.: Conceptualization, data curation, formal analysis, validation, investigation, visualization, methodology, and writing–review and editing; L.S.M.: Formal analysis, validation, investigation, visualization, methodology, and writing–review and editing; Y.W.: formal analysis, visualization, methodology, and writing–review and editing; E.B.: formal analysis, visualization, and validation; S.U.: formal analysis, investigation, visualization, and methodology; E.S.: formal analysis, investigation, and validation; S.K.D.: formal analysis, investigation, visualization, and validation; C.I.: Data curation and formal analysis; S.P.: investigation and formal analysis, W.Y.: investigation and formal analysis; C.L.: investigation, formal analysis, visualization, and methodology; N.B. J.: investigation, formal analysis, and review and editing; V.V.: formal analysis; W.H.: formal analysis and validation; A.C.-R.: formal analysis; Z.K.: investigation and formal analysis; H.D.: investigation and formal analysis; W.X.: investigation and formal analysis; H.-J.C.: investigation, formal analysis, validation, investigation, and visualization; M.H.: formal analysis; T.K.: formal analysis, investigation, and methodology; C.-A.M.: Investigation and methodology; R.A.-F.: formal analysis and validation; M.J.B.: Formal analysis and validation; J.L.: Formal analysis and validation; N.Z.: investigation, resources, supervision, and methodology; G.L.-B.: Conceptualization, resources, and supervision; V.d.F.: Conceptualization, resources, and supervision; Z.A.: Conceptualization, validation, investigation, visualization, methodology; A.K.S.: Conceptualization, resources, data curation, formal analysis, supervision, funding acquisition, validation, investigation, visualization, methodology, project administration, and review and editing.

## Competing interests

A.K.S.: Consulting (Merck, Kiyatec, AstraZeneca, Onxeo, Iylon, ImmunoGen, GSK), shareholder (BioPath), research support (M-Trap). N.Z., Z.A., A.K.S., Z.K., and H.D. are inventors for U.S. patent No. 63/004,149 'CD5L binding antibodies and uses for the same priority claim' filed by The University of Texas Systems. The remaining authors declare no competing interests.

## Additional information

[1]Department of Gynecologic Oncology and Reproductive Medicine, The University of Texas MD Anderson Cancer Center, Houston, Texas 77030, USA. [2]Department of Experimental Therapeutics, The University of Texas MD Anderson Cancer Center, Houston, TX 77030, USA. [3]Istituto di Endocrinologia ed Oncologia Sperimentale, CNR, Naples, Italy. [4]Laboratory of Disease Modeling and Therapeutics, Korea Research Institute of Bioscience and Biotechnology, Daejeon, Republic of Korea. [5]Center for RNA Interference and Non-Coding RNA, The University of Texas MD Anderson Cancer Center, Houston, TX 77030, USA. [6]Department of Obstetrics and Gynecology, Medical College of Wisconsin, Milwaukee, WI 53226, USA. [7]Department of Molecular & Cellular Oncology, The University of Texas MD Anderson Cancer Center, Houston, TX 77030, USA. [8]Wave Life Sciences, 733 Concord Avenue, Cambridge, MA 02138, USA. [9]Department of Genetic Medicines, Alloy Therapeutics, Waltham, USA. [10]Department of Obstetrics and Gynecology, Ludwig Maximilians University of Munich, Munich, Germany. [11]German Cancer Consortium (DKTK), German Cancer Research Center, Munich, Germany. [12]Texas Therapeutics Institute, Brown Foundation Institute of Molecular Medicine, The University of Texas Health Science Center at Houston, Houston, TX 77030, USA. [13]Department of Obstetrics and Gynecology, Chung-Ang University, College of Medicine, Seoul, Republic of Korea. [14]Department of Obstetrics and Gynecology, Chung-Ang University Gwangmyeong Hospital, College of Medicine Chung-Ang University, Seoul, South Korea. [15]CPRIT Single Core, Department of Genetics, The University of Texas MD Anderson Cancer Center, Houston, TX 77030, USA. [16]Ruiz Department of Ophthalmology and Visual Science, McGovern Medical School at The University of Texas Health Science Center at Houston (UTHealth), Houston, TX 77030, USA. [17]The MD Anderson Cancer Center/UTHealth Graduate School of Biomedical Sciences, Houston, TX 77030, USA. [18]Department of Pathology, Wayne State University, Detroit, MI 48201, USA. [19]Winthrop P. Rockefeller Cancer Institute at the University of Arkansas for Medical Sciences, Little Rock, AR, USA. [20]Department of Pathology, The University of Texas MD Anderson Cancer Center, Houston, TX 77030, USA. [21]National Research Council (CNR), Institute of Genetic and Biomedical Research (IRGB)-UOS Milan via Rita Levi Montalcini, 20090 Pieve Emanuele, MI, Italy. [22]Harvard Medical School Initiative for RNA Medicine, Harvard Medical School, Boston, MA 02115, USA. [23]These authors contributed equally: Christopher J. LaFargue, Paola Amero, Kyunghee Noh. ✉e-mail: ywen2@mdanderson.org; asood@mdanderson.org

