## [Peer Review File · Nature Communications]

Overcoming adaptive resistance to anti-VEGF therapy by targeting CD5LREVIEWER COMMENTS

Reviewer #1 (Remarks to the Author):

LaFargue et al., describe a novel mechanism that contributes to resistance against antiangiogenic (anti-VEGF) therapy. Anti-VEGF treatment in ovarian cancer mouse model led to a profound upregulation of CD5L. This was mediated by hypoxia-driven PPAR γ activation. CD5L appears to have some signaling function leading to AKT phosphorylation and CD36 expression. Lastly, and this is by far the strongest aspect of this manuscript, the authors show that blocking CD5L with antibodies or aptamers blocks resistance to anti-VEGF therapy.

The topic is of outstanding interest as the large majority of patients treated with anti-VEGF drugs develops resistance. The manuscript is well written.

Unfortunately, the use of cell line instead of primary cells the poor phenotypic characterization of mouse models and novel therapeutic substances and the poor evaluation of the signaling mechanism very much limits my enthusiasm about this manuscript.

Major points:

It is surprising that the authors performed in vitro experiments with the RF24 endothelial cell line instead of primary endothelial cells. This would allow them to also study capillary formation using spheroid or bead methods. In addition, it would be interesting to determine in a much more thorough manner which endothelial cells express CD5L under physiological conditions (arterial, capillary, venous, continuous, discontinuous etc.). Is CD5L expression different between proliferation and resting endothelial cells (e.g. in the postnatal retina)?

Also, it remained somehow unclear which cell type is producing secreted CD5L particularly in tumors.

The analysis of PPAR γ deficient mice gives interesting results (fig. 4). However, it remains completely unclear whether this is mediated by CD5L downstream of PPAR γ . It is well known that PPAR γ is a master regulator of cell metabolism and regulates expression of numerous genes. Angiogenesis is strongly dependent on glucose and fatty acid oxidation (see seminal work by the Carmeliet group). In addition, the authors do not mention that these Tie2-Cre; flox-PPAR γ mice exhibit additional vascular problems that may contribute to the tumor progression phenotype.

The CD5L signaling mechanism through CD36 is not very well analyzed. The authors could test the importance in EC-specific CD36-deficient mice. Also the downstream events are elusive. PI3K/AKT phosphorylation is involved in angiogenesis but this signaling mode can also be activated by numerous other angiogenic factors besides of CD5L. Again, one could also speculate that altered CD36 expression would alter the amount of fatty acid uptake in endothelial cells which could substantially impact on cell metabolism and angiogenesis.

The CD5L blocking substances show impressive effects. Unfortunately, the authors do not show any data how well these are tolerated by mice. This is not acceptable. Moreover, how do these drugs affect endothelial cells in non-tumor models. Is there a general effect on angiogenesis in mice?

Reviewer #2 (Remarks to the Author):

The manuscript of LaFargue et al demonstrates the interesting observation that CD5 antigen-like precursor (CD5L) is upregulated in response to anti-angiogenic therapy like bevacizumab (anti-VEGF monoclonal antibody) leading to the emergence of adaptive resistance. This observation makes CD5L an interesting target to reverse the resistant phenotype after/during anti-VEGF therapy. They have developed novel antibody and aptamers to CD5L to prove this effect in vivo, in elegant orthoptic models for ovarian cancer. In general I think the findings are novel and definitely of interest to others in the community and broader.

However, I have a few concerns:

- The authors elegantly show that PPAR- γ serves as an upstream regulator of CD5L, demonstrated with over-expression and siRNA technology. Why did they not use CRISPR/Cas9 to fully knock out PPAR- γ ? Also for CD5L they use knock down and not knock out technology, which would give a more

clear-cut result.

- As a tumormodel, the authors use an SKOV3ip1 ovarian cancer mouse model and BLI to follow the outgrowth in time (figure 1). Later, they use murine ID8 ovarian cancer cells, injected intraperitoneally in C57BL/6 mice. Now they do an endpoint analyses of the size of the tumors (fig 4), why was also here not the luciferase and BLI measurement used, to follow outgrowth in time and have more (quantative) data of the experiments? The endpoint seems to be chosen a bit arbitrary and more data could have been collected with less mice. Same is true for figure 5.

Nonetheless, this manuscript contains many interesting data with proper conclusions, and I would recommend publication.

Reviewer #3 (Remarks to the Author):

This is a comprehensive investigation into the molecular mechanisms of acquired resistance to anti-VEGF therapy. The authors took a straightforward and effective approach to discovering that CD5L mediates this resistance and blocking CD5L is a potential therapeutic strategy to restore sensitivity. This is an important problem in ovarian cancer therapy and other cancers where anti-VEGF therapy is frequently used clinically.

I have only minor comments:

1. Please comment on whether PPAR γ and HIF1 α were found to be upregulated in the initial profiling experiment shown in Figure 1A. (If they were not, then this could possibly be due to the timing of the samples chosen.)
2. In Figure 2L, it would be useful to know whether the YC1 causes decrease in PPAR γ and CD5L under hypoxic conditions. Please clarify the conditions of this experiment.
3. For better flow of ideas, consider putting all of Figure 7 earlier in the results. I was wondering the whole time whether the CD5L was only a lab phenomenon, or if it occurred in humans. This is purely stylistic and I would leave this up to authors discretion.

Reviewer #4 (Remarks to the Author):

The manuscript by LaFargue et al. introduced CD5L as a new target for anti-angiogenic treatment targeting the vascular endothelial growth factor (VEGF) pathway, especially for the models with drug-resistance. The authors identified CD5L as an upregulated protein for the cancer with adaptive resistance led by anti-angiogenic therapy, then a series of studies have been performed. They have demonstrated that the upregulation of CD5L is caused by the hypoxia-induced PPAR- γ overexpression and the increase of PI3K/AKT expression levels caused by the exogenous CD5L. Then, they developed novel antibodies and aptamers targeting CD5L and obtained great therapeutic effects for cancer cells with drug resistance and the mouse models carried drug resistance tumors. The manuscript is well-organized, and the authors have provided well-designed in vitro and in vivo experiments and abundant solid data for this important topic. It should be accepted for publication after the following minor issues are well addressed:

1. In the first paragraph of Introduction, the authors mentioned that "While many patients benefit from such therapies, virtually all patients will eventually develop relapse or progression of disease." Could the authors provide several more references to support this statement?
2. In the first paragraph on page 12, the authors mentioned that "To assess the functional effects of S76.T-aptamer, we evaluated pAKT expression in RF24 resistant cells following treatment with S76.T." It is not clear what drug/antibody does this resistant cell line has resistance to (B20 or bevacizumab or maybe another drug)?
3. For the results in Figure 6G, H, I and J, is it possible that the authors can add the data for aptamer S76.T only to compare with S76.T plus B20 antibody? Besides, can the authors discuss about whether the aptamer itself can be used as a therapeutic agent or it should be used with the VEGF antibodies, like B20 or bevacizumab (from the Figure 6D , great cancer killing result can be achieved only using aptamer)?

Reviewer comments:

Reviewer #1 (Remarks to the Author):

LaFargue et al., describe a novel mechanism that contributes to resistance against antiangiogenic (anti-VEGF) therapy. Anti-VEGF treatment in ovarian cancer mouse model led to a profound upregulation of CD5L. This was mediated by hypoxia-driven PPAR γ activation. CD5L appears to have some signaling function leading to AKT phosphorylation and CD36 expression. Lastly, and this is by far the strongest aspect of this manuscript, the authors show that blocking CD5L with antibodies or aptamers blocks resistance to anti-VEGF therapy.

The topic is of outstanding interest as the large majority of patients treated with anti-VEGF drugs develops resistance. The manuscript is well written. Unfortunately, the use of cell line instead of primary cells the poor phenotypic characterization of mouse models and novel therapeutic substances and the poor evaluation of the signaling mechanism very much limits my enthusiasm about this manuscript. Major points:

1. It is surprising that the authors performed in vitro experiments with the RF24 endothelial cell line instead of primary endothelial cells. This would allow them to also study capillary formation using spheroid or bead methods. In addition, it would be interesting to determine in a much more thorough manner which endothelial cells express CD5L under physiological conditions (arterial, capillary, venous, continuous, discontinuous etc.). Is CD5L expression different between proliferation and resting endothelial cells (e.g. in the postnatal retina)?

Response: As suggested, we have added data with two primary endothelial cells, which include human pulmonary artery endothelial cells (HPAEC) and human umbilical venous endothelial cells (HUVEC). As shown below, treatment of these cells with CD5L recombinant protein resulted in increased tube formation. Addition of CD5L specific antibody (R-35 antibody) to endothelial cells treated with CD5L protein blocked the observed increase in tube formation when cells were treated with control antibody plus CD5L protein (A: HPAEC and B: HUVEC).

Next, we performed capillary formation assays using spheroids. As shown below, treatment of endothelial cells (HUVEC) with CD5L recombinant protein resulted in increased capillary formation and sprout length. Addition of R-35 antibody to HUVEC spheroids treated with CD5L protein blocked the observed increase in sprout length in HUVEC spheroids induced by control antibody plus CD5L protein (C). This data is now included in Figure S8.

Effect of CD5L on tube formation and capillary formation

2. Is CD5L expression different between proliferation and resting endothelial cells (e.g. in the postnatal retina)?

Response: We appreciate this comment. As the reviewer mentioned, we checked CD5L expression in mouse retinal endothelial cells during postnatal development from publicly available expression data (GEO; GSE86788). We found that CD5L expression is higher at days P6, P10 and P50 compared to days P15 and P21. Since this data is about whole retina, we also checked the expression of CD5L in proliferating and resting arterial and venous endothelial cells. The endothelial cells were serum-starved for 72h (resting); then, we compared CD5L expression in these with proliferating endothelial cells. As shown below, we did not find any significant difference in CD5L expression in proliferating (PEC) versus resting (QEC) conditions of HPAEC (A) and HUVEC (B). We also checked the expression of quiescence marker p27 (cyclin dependent kinase inhibitor). As shown below, increased expression of p27 in both cells further proves that cells were in resting condition.

Carmeliet et al. have previously reported that quiescent endothelial cells (QEC) reprogram their metabolism to enhance redox homeostasis¹. They isolated HUVECs and HUAECs from umbilical cords obtained from multiple donors. We checked CD5L expression in proliferating and quiescent endothelial cells from multiple donors using publicly available expression data (GEO; GSE89174) and found no difference in CD5L expression between PEC and QEC.

Expression of CD5L in proliferating and resting endothelial cells

3. Also, it remained somehow unclear which cell type is producing secreted CD5L particularly in tumors.

Response: We appreciate this important comment. To address this comment, we analyzed 8 high-grade serous ovarian cancer samples using single cell RNA sequencing of six populations including T cells, monocytes, epithelial cells, fibroblasts, natural killer cells and B cells. As shown below, we observed almost no expression of CD5L in any of these populations, with only a few monocytes and B-cells showing some level of expression (each dot represents one single cell). This data is now included in Figure S1.

Expression of CD5L in human high-grade serous ovarian tumors

4. The analysis of PPAR γ deficient mice gives interesting results (fig. 4). However, it remains completely unclear whether this is mediated by CD5L downstream of PPAR γ . It is well known that PPAR γ is a master regulator of cell metabolism and regulates expression of numerous genes. Angiogenesis is strongly dependent on glucose and fatty acid oxidation (see seminal work by the Carmeliet group). In addition, the authors do not mention that these Tie2-Cre; flox-PPAR γ mice exhibit additional vascular problems that may contribute to the tumor progression phenotype.

Response: We appreciate this important point. As mentioned by the reviewer, PPARG can regulate the expression of various genes involved in lipid metabolism. Compared to WT mice, *PPARG* KO mice fed with standard low fat diet did not show any significant difference in body weight, liver/body weight, fat content, fasting glucose, TG and FFA levels; whereas significant differences were observed in animals given high-fatty acid diet^{2,3}. To investigate the role of PPARG in regulating CD5L downstream effects, we performed knockout of PPARG in RF24 endothelial cells using CRISPR/Cas9 and tested the effects on CD5L-induced effects on angiogenesis (e.g., tube formation). The effect of CRISPR/Cas9 knockout of PPARG was validated by Western blots. *PPARG* knockout resulted in less tube formation compared to scramble control treated cells. Importantly, addition of CD5L recombinant protein to these cells rescued the decreased tube formation under *PPARG* knockout, as shown below. This data is now included in Figure S7.

Effect of PPARG silencing on angiogenesis

5. The CD5L signaling mechanism through CD36 is not very well analyzed. The authors could test the importance in EC-specific CD36- deficient mice. Also the downstream events are elusive. PI3K/AKT phosphorylation is involved in angiogenesis but this signaling mode can also be activated by numerous other angiogenic factors besides of CD5L. Again, one could also speculate that altered CD36 expression would alter the amount of fatty acid uptake in endothelial cells which could substantially impact on cell metabolism and angiogenesis.

Response: Thank you for your comments. As suggested, we obtained the endothelial-specific CD36 knockout mouse model (CD36^{flox/flox}Tie2^{Cre})⁴. Briefly, luciferase labelled murine ovarian cancer ID8-Luc cells (1.0×10^6 cells) were inoculated into the peritoneal cavity of age-matched (4 to 6 weeks old), female C57/BL6 mice (Taconic Biosciences), and CD36^{flox/flox}Tie2^{Cre} mice. After the initial establishment of tumors, as measured by IVIS imaging (1×10^4 photons/second/cm²/sr) at day 8, mice were treated with either anti-CD5L antibody (R35) to 1) WT (n=10), 2) CD36^{flox/flox}Tie2^{Cre} (n=8) groups; or control IgG to group 3) CD36^{flox/flox}Tie2^{Cre} (n=8) for four weeks. Tumor growth was monitored by weekly bioluminescence imaging. Animals were sacrificed when tumor burden exceeded the established guidelines by IACUC (~35 days post-inoculation) (A). The inhibitory effects of R35 antibody depended on presence of endothelial CD36. Mice

with tissue specific knockout of endothelial CD36 did not respond well to the R35 antibody (B and D) compared to WT C57/BL6 mice. The mouse weights of each group were similar (Figure S7C) across the three groups. The growth and progression of syngeneic ID8-luc tumors in each group were monitored by bioluminescence imaging (E and F). This data now included in Figure S10.

In vivo effects of anti-CD5L-antibody (R35) in syngeneic murine ID8 ovarian cancer model

Next, we checked the effects of the R35 antibody in endothelial specific CD36 knockout tumors by staining the tumor tissue with CD31 and pAKT antibodies. As shown below, no difference was observed between CD36 endothelial specific KO mice treated with either R35 antibody or IgG compared to WT-R35 treated mice. This data now included in Figure S11.

Effects of anti-CD5L antibody (R-35) on angiogenesis in endothelial-specific *CD36* knockout ID8 tumors

To address if altered *CD36* expression could change the amount of fatty acid (FA) uptake in endothelial cells, we tested exogenous FA uptake capacity in RF24 cells with or without *CD36* knockdown. Results indicate that *CD36* knockdown reversed the enhancement of FA uptake induced by *CD5L* protein (r*CD5L*). This data is now included in Figure S12.

Fatty acid uptake in RF24 cells with *CD5L* stimulation or *CD36* knockdown

6. The *CD5L* blocking substances show impressive effects. Unfortunately, the authors do not show any data how well these are tolerated by mice. This is not acceptable. Moreover, how do these drugs affect endothelial cells in non-tumor models. Is there a general effect on angiogenesis in mice?

Response: As suggested, we examined the effects of *CD5L* (R-35) on normal C57/BL6 mice. Animals were treated with either control IgG or R-35 for two weeks (once per week). Blood was collected and CBC (complete blood count) and other analyses were performed. As shown below in panel A, there was no significant difference in WBC, hemoglobin, or platelet counts. Moreover, there were no differences in serum ALT, AST and LDH levels between R-35 and control IgG treated mice (values are means \pm standard error, $n = 4$ (two-tailed t test). H&E staining of multiple organs, including lung, liver, kidney, and spleen from R-35 treated mice were examined by two gynecologic pathologists and did not show any difference in

histopathological findings compared to control IgG treated animals (B). To check how R-35 antibody affects endothelial cells in non-tumor models, we stained blood vessels of different organs using a CD31 antibody and did not observe any difference between the control IgG or R-35 Ab treated groups, suggesting that CD5L blocking antibodies did not affect angiogenesis in normal organs (C). This data is now included in Figure S9.

Effect of CD5L blocking antibody R-35 on normal mice

Reviewer #2 (Remarks to the Author):

The manuscript of LaFargue et al demonstrates the interesting observation that CD5 antigen-like precursor (CD5L) is upregulated in response to anti-angiogenic therapy like bevacizumab (anti-VEGF monoclonal antibody) leading to the emergence of adaptive resistance. This observation makes CD5L an interesting target to reverse the resistant phenotype after/during anti-VEGF therapy. They have developed novel antibody and aptamers to CD5L to prove this effect in vivo, in elegant orthoptic models for ovarian cancer. In general I think the findings are novel and definitely of interest to others in the community and broader. However, I have a few concerns:

1. The authors elegantly show that PPAR- γ serves as an upstream regulator of CD5L, demonstrated with over-expression and siRNA technology. Why did they not use CRISPR/Cas9 to fully knock out PPAR- γ ? Also for CD5L they use knock down and not knock out technology, which would give a more clear-cut result.
Response: As suggested, we used CRISPR/Cas9 to knock out CD5L in RF24 endothelial cells, and the effects of CRISPR/Cas9 knockout of *PPARG* or *CD5L* on tube formation of RF24 cells were tested. As shown below, CD5L K/O cells formed fewer tubes compared to scramble control cells. Importantly, addition of CD5L recombinant protein to these cells rescued the decreased tube formation induced by CD5L knockout. This data is now included in Figure S2.

Effect of CD5L silencing on angiogenesis

2. As a tumor model, the authors use an SKOV3ip1 ovarian cancer mouse model and BLI to follow the outgrowth in time (figure 1). Later, they use murine ID8 ovarian cancer cells, injected intraperitoneally in C57BL/6 mice. Now they do an endpoint analyses of the size of the tumors (fig 4), why was also here not the luciferase and BLI measurement used, to follow outgrowth in time and have more (quantitative) data of the experiments? The endpoint seems to be chosen a bit arbitrary and more data could have been collected with less mice. Same is true for figure 5.

Response: In Figure 1, we used luciferase labeled ovarian cancer cells (SKOV3ip1) to separate the mice into sensitive and resistant groups based on the bioluminescence signal. For Figure 4A to 4C, the aim of the experiment was to evaluate tumor growth and the experiment was terminated when mice became moribund in any group due to high tumor burden and/or ascites. Moreover, ID8 tumor bearing mice have more ascites, which can interfere with the luminescence signal for tumor. For Figure 4F (survival), the percent survival was based on animals becoming moribund individually and luciferase measurement was not needed for this end-point. For Figure 5, we used the SKOV3ip1 ovarian cancer mouse model and the end-point was to evaluate tumor growth. Animals were sacrificed and tumors were harvested when mice became sick due to tumor burden in any group. This information has now been clarified in the Method section.

3. Nonetheless, this manuscript contains many interesting data with proper conclusions, and I would recommend publication.

Response: We thank the reviewer for the positive comment.

Reviewer #3 (Remarks to the Author):

This is a comprehensive investigation into the molecular mechanisms of acquired resistance to anti-VEGF therapy. The authors took a straightforward and effective approach to discovering that CD5L mediates this resistance and blocking CD5L is a potential therapeutic strategy to restore sensitivity. This is an important problem in ovarian cancer therapy and other cancers where anti-VEGF therapy is frequently used clinically.

I have only minor comments:

1. Please comment on whether PPAR γ and HIF1 α were found to be upregulated in the initial profiling experiment shown in Figure 1A. (If they were not, then this could possibly be due to the timing of the samples chosen.)

Response: Both PPAR γ and HIF1 α expression levels were higher (the difference was significant for HIF1 α) in resistant endothelial cells compared with sensitive endothelial cells. We have now added this information to Figure S5A.

2. In Figure 2L, it would be useful to know whether the YC1 causes decrease in PPAR γ and CD5L under hypoxic conditions. Please clarify the conditions of this experiment.

Response: YC1 causes decrease in both PPAR γ and CD5L levels under hypoxic conditions. This data has been added to Figure S5B and we have now clarified the conditions in the Results and Methods sections. In addition, we also examined the expression of PPAR γ and CD5L in the presence of HIF1 α siRNA. As shown in panel B below, HIF1 α siRNA treated endothelial cells have decreased expression of both PPAR γ and CD5L compared to control siRNA treated cells. This data is now included in Figure S5B.

Upregulation of PPAR γ and HIF1 α in anti-VEGF antibody-resistant tumor endothelial cells

3. For better flow of ideas, consider putting all of Figure 7 earlier in the results. I was wondering the whole time whether the CD5L was only a lab phenomenon, or if it occurred in humans. This is purely stylistic and I would leave this up to authors discretion.

Response: We appreciate the reviewer's suggestion. We would respectfully like to retain Figure 7 as the final figure as we think it will be easier for the readers to follow the flow of the preclinical models prior to presenting this data.

Reviewer #4 (Remarks to the Author):

The manuscript by LaFargue et al. introduced CD5L as a new target for anti-angiogenic treatment targeting the vascular endothelial growth factor (VEGF) pathway, especially for the models with drug-resistance. The authors identified CD5L as an upregulated protein for the cancer with adaptive resistance led by anti-angiogenic therapy, then a series of studies have been performed. They have demonstrated that the

upregulation of CD5L is caused by the hypoxia-induced PPAR- γ overexpression and the increase of PI3K/AKT expression levels caused by the exogenous CD5L. Then, they developed novel antibodies and aptamers targeting CD5L and obtained great therapeutic effects for cancer cells with drug resistance and the mouse models carried drug resistance tumors. The manuscript is well-organized, and the authors have provided well-designed in vitro and in vivo experiments and abundant solid data for this important topic. It should be accepted for publication after the following minor issues are well addressed:

1. In the first paragraph of Introduction, the authors mentioned that “While many patients benefit from such therapies, virtually all patients will eventually develop relapse or progression of disease.” Could the authors provide several more references to support this statement?

Response: As suggested by reviewer, we have now included additional references in the Introduction.

- a. Ellis, LM and Hicklin DJ. Pathways mediating resistance to vascular endothelial growth factor-targeted therapy. *Clin. Cancer Res.*, 14 (2008), pp. 6371-6375
- b. Kerbel RS. Tumor angiogenesis. *N. Engl. J. Med.*, 358 (2008), pp. 2039-2049
- c. Shojaei, F and Ferrara, N. Role of the microenvironment in tumor growth and in refractoriness/resistance to anti-angiogenic therapies. *Drug Resist. Updat.*, 11 (2008), pp. 219-230

2. In the first paragraph on page 12, the authors mentioned that “To assess the functional effects of S76.T-aptamer, we evaluated pAKT expression in RF24 resistant cells following treatment with S76.T.” It is not clear what drug/antibody does this resistant cell line has resistance to (B20 or bevacizumab or maybe another drug)?

Response: As suggested, we have now clarified this information in the manuscript; specifically, we used bevacizumab.

3. For the results in Figure 6G, H, I and J, is it possible that the authors can add the data for aptamer S76.T only to compare with S76.T plus B20 antibody? Besides, can the authors discuss about whether the aptamer itself can be used as a therapeutic agent or it should be used with the VEGF antibodies, like B20 or bevacizumab (from the Figure 6D, great cancer killing result can be achieved only using aptamer)?

Response: As suggested, we have now added the data for S76.T aptamer alone. As shown in panel A below, mice treated with aptamer alone showed reduction in tumor burden; whereas combination of aptamer and B20 treatment showed greater reduction in tumor weight, tumor nodules (Figure B to D), angiogenesis (E) and proliferation (F) compared to scramble aptamer + IgG treated mice. The aptamer alone could potentially be used as a therapeutic agent, but it is likely that better efficacy would be observed when combined with a VEGF antibody. This data is now included in Figure 6.

CD5L targeted aptamer (S76.T) blocks resistance to anti-VEGF therapy.

References:

- 1 Kalucka, J. *et al.* Quiescent Endothelial Cells Upregulate Fatty Acid beta-Oxidation for Vasculoprotection via Redox Homeostasis. *Cell metabolism* **28**, 881-894 e813, doi:10.1016/j.cmet.2018.07.016 (2018).
- 2 Kanda, T. *et al.* PPARgamma in the endothelium regulates metabolic responses to high-fat diet in mice. *The Journal of clinical investigation* **119**, 110-124, doi:10.1172/JCI36233 (2009).
- 3 Guignabert, C. *et al.* Tie2-mediated loss of peroxisome proliferator-activated receptor-gamma in mice causes PDGF receptor-beta-dependent pulmonary arterial muscularization. *American journal of physiology. Lung cellular and molecular physiology* **297**, L1082-1090, doi:10.1152/ajplung.00199.2009 (2009).
- 4 Son, N. H. *et al.* Endothelial cell CD36 optimizes tissue fatty acid uptake. *The Journal of clinical investigation* **128**, 4329-4342, doi:10.1172/JCI99315 (2018).
- 5 Bou Khzam, L., Son, N. H., Mullick, A. E., Abumrad, N. A. & Goldberg, I. J. Endothelial cell CD36 deficiency prevents normal angiogenesis and vascular repair. *American journal of translational research* **12**, 7737-7761 (2020).
- 6 Feng, W. W. *et al.* CD36-Mediated Metabolic Rewiring of Breast Cancer Cells Promotes Resistance to HER2-Targeted Therapies. *Cell reports* **29**, 3405-3420 e3405, doi:10.1016/j.celrep.2019.11.008 (2019).

REVIEWER COMMENTS

Reviewer #1 (Remarks to the Author):

The authors have substantially improved this manuscript in several parts. Nevertheless, some of my questions have not been addressed. Therefore I am repeating these:

These issues remain:

- 1) to determine in a much more thorough manner which endothelial cells express CD5L under physiological conditions (arterial, capillary, venous, continuous, discontinuous etc.).
- 2) Please provide CD5L staining of the postnatal mouse retina including endothelial markers to at least determine the expression in tip vs. stalk cells and arterial vs. venous cells.
- 3) The authors show that several cell types do NOT express CD5L in tumors. So which cell type does express it?

Reviewer #2 (Remarks to the Author):

The revised manuscript has improved a lot, and the comments were sufficiently addressed.

Reviewer #3 (Remarks to the Author):

The authors have thoroughly responded to my comments and I have no further concerns.

Reviewer #4 (Remarks to the Author):

The authors have answered the questions from the reviewers well, therefore this manuscript should be accepted by Nature Communications.

Reviewer comments:

Reviewer #1 (Remarks to the Author):

The authors have substantially improved this manuscript in several parts. Nevertheless, some of my questions have not been addressed. Therefore, I am repeating these:

These issues remain:

1) to determine in a much more thorough manner which endothelial cells express CD5L under physiological conditions (arterial, capillary, venous, continuous, discontinuous etc.).

Response: We appreciate this comment. As suggested by the reviewer, we examined the expression of CD5L in different endothelial cell populations under physiological conditions including liver, lung and kidney. These tissues were harvested from healthy mice (C57BL6, ~6 weeks of age). Liver tissue was stained with CD31 and CD5L antibodies to show the expression of CD5L in hepatic arterial, portal venous and discontinuous endothelial (or sinusoid) cells (Shetty, Lalor et al. 2018). As shown below, hepatic arterial (yellow arrow), portal venous (PV) and discontinuous endothelium (white arrows) showed weak expression of CD5L (scale bar = 50 μ m).

CD5L expression in physiological conditions from mouse portal venous, hepatic arterial and discontinuous endothelial cells

To assess the expression of CD5L in continuous endothelial cells, we stained mouse lung tissue with CD31 and CD5L antibodies. As shown below, continuous endothelium shows modest expression of CD5L (denoted by white arrows; scale bar = 50 μ m).

CD5L expression in mouse lung continuous endothelial cells

To assess CD5L expression in capillary endothelial cells, we stained normal mouse kidney tissue with CD5L and CD31 antibodies. Normal kidneys have peritubular capillaries surrounding tubules (Kida 2020). As shown below, few capillary endothelial cells showed modest CD5L expression (denoted by white arrows; scale bar = 50 μ m). As requested, these data have now been incorporated into Supplementary Figure 9A.

CD5L expression in mouse kidney capillary endothelial cells

2) Please provide CD5L staining of the postnatal mouse retina including endothelial markers to at least determine the expression in tip vs. stalk cells and arterial vs. venous cells.

Response: As requested, we checked the expression of CD5L in mouse pup retinal endothelial cells during postnatal development (P5). We stained mouse pup retina with CD5L and CD31 antibodies; we did not observe specific co-localization in stalk and tip cells (A). However, we did see some specific staining in established blood vessels (B; scale bar = 50 μ m).

A

CD5L expression in mouse pup retinal tip and stalk endothelial cells (p5)

B

CD5L expression in mouse pup retinal endothelial cells (p5)

3) The authors show that several cell types do NOT express CD5L in tumors. So which cell type does express it?

Response: Thanks for the comment. Macrophages are known to express CD5L (Sanjurjo, Aran et al. 2015); to check whether tumor macrophages express CD5L, we stained SKOV3ip1 tumor tissue with F4/80

(macrophage marker) and CD5L antibodies. As shown below, macrophages from SKOV3 tumors indeed express CD5L (denoted by white arrows; scale bar = 20µm). In the manuscript, we also observed the expression of CD5L in tumor-associated endothelial cells (please see Figure 1C & 7C).

CD5L expression in SKOV3ip1 tumor macrophages

Reviewer #2 (Remarks to the Author):

The revised manuscript has improved a lot, and the comments were sufficiently addressed.

Response: We thank the reviewer for this input.

Reviewer #3 (Remarks to the Author):

The authors have thoroughly responded to my comments and I have no further concerns.

Response: We thank the reviewer for this input.

Reviewer #4 (Remarks to the Author):

The authors have answered the questions from the reviewers well, therefore this manuscript should be accepted by Nature Communications.

Response: We thank the reviewer for this input.

References

Kida, Y. (2020). "Peritubular Capillary Rarefaction: An Underappreciated Regulator of CKD Progression." Int J Mol Sci **21**(21).

Sanjurjo, L., G. Aran, N. Roher, A. F. Valledor and M. R. Sarrias (2015). "AIM/CD5L: a key protein in the control of immune homeostasis and inflammatory disease." J Leukoc Biol **98**(2): 173-184.

Shetty, S., P. F. Lalor and D. H. Adams (2018). "Liver sinusoidal endothelial cells - gatekeepers of hepatic immunity." Nat Rev Gastroenterol Hepatol **15**(9): 555-567.

REVIEWER COMMENTS

Reviewer #1 (Remarks to the Author):

The authors have addressed all of my concerns. There are no further questions.

Reviewer #5 (Remarks to the Author):

General: In this manuscript, titled "Overcoming adaptive resistance to anti-VEGF therapy by targeting CD5L" the authors propose that CD5 antigen like precursor is upregulated in response to anti-angiogenic therapy, which leads to adaptive resistance. They show that an RNA-aptamer or a monoclonal antibody against CD5L reduces its pro-angiogenic activity in vitro and in vivo settings and suggest that anti-CD5L strategy might be a clinically feasible approach to improve the efficacy of the currently used anti-angiogenic therapy. I understood that this manuscript went already through one round of revisions and will therefore only comment on single cell RNA-seq experiment that the authors included in the revised version.

Major comments:

1. Single cell RNA-seq data are clearly a minor part of the study, with all results included in Supplementary figure 1. While I appreciate that the authors do not attempt to inflate or overemphasize this data, the current presentation is too simplified. Additional data and description will need to be included in the manuscript if this part is to stay. At minimum, the authors should include the following:
 - a. Method for single cell RNA-seq data. Currently, only the sample preparation procedure is described, but no information is provided for the transcriptomics data analysis.
 - b. The authors should indicate how a quality check was performed on their scRNA-seq data – at minimum what were the constraints they used to distinguish a high-quality cell (minimum/maximum number of genes per cell, percentage of mitochondrial genes), how many cells per sample were included in the analysis for each sample.
 - c. A t-SNE or UMAP visualizing all the data should be included in figure S1.
 - d. Clustering of the data should be included to visualize how the cell types were detected.
2. In figure S1 the authors claim that they "observed almost no expression of CD5L" in any of the six populations including T cells, monocytes, epithelial cells, fibroblasts, natural killer cells, and B cells. Why were these cell types selected? Expression in endothelial cells must be included in this graph for comparison. To exclude that there is another population that is an important source of CD5L, expression in all clusters should be included in the graph.
3. Another important point is that the scRNA-seq was performed in human samples. However, in the text the authors use this data to support a selective CD5L expression in endothelial cells in a mouse. A question therefore remains what the expression of CD5L in murine tumor cell types is. Based on Figure S1 it cannot be concluded that other cell types in the murine tumors are irrelevant as a source of CD5L.

Reviewer comments:

Reviewer #5 (Remarks to the Author):

Major comments:

1. Single cell RNA-seq data are clearly a minor part of the study, with all results included in Supplementary figure 1. While I appreciate that the authors do not attempt to inflate or overemphasize this data, the current presentation is too simplified. Additional data and description will need to be included in the manuscript if this part is to stay. At minimum, the authors should include the following:
a. Method for single cell RNA-seq data. Currently, only the sample preparation procedure is described, but no information is provided for the transcriptomics data analysis.

Response: Our data were analyzed using Cell Ranger software version 3.1.0, and a gene expression matrix based on universal molecular identifier (UMI) counts was generated. We further employed Seurat version 3.1.4 for unbiased cell clustering analysis. Briefly, we first filtered the matrix based on minimum/maximum cut-offs for genes/cell, cells/gene, and optional parameters such as mitochondrial gene UMI count as a percentage of total and normalized the data. Next, the variable genes were identified and scaled the data. Thirdly, we identified principal components; and finally identified cell “neighbors” and cell clusters. This information was also added in the Protocol Exchange.

b. The authors should indicate how a quality check was performed on their scRNA-seq data – at minimum what were the constraints they used to distinguish a high-quality cell (minimum/maximum number of genes per cell, percentage of mitochondrial genes), how many cells per sample were included in the analysis for each sample.

Response: We defined the percentage of mitochondrial genes should be less than 20 and minimum number of genes per cell should be larger than 200. The table below shows the number of cells per sample that were included in this analysis.

Table: Number of cells per sample

Sample	Cell number
N1	3931
OMT-1.1	6720
OMT-1.2	1049
OMT-3.1	1854
OMT-3.2	5133
T1	3100
T6	3508
T6.1	2344

c. A t-SNE or UMAP visualizing all the data should be included in figure S1.

Response: We thank the reviewer and have now included the UMAP for all the data in Figure S1, as shown below (Figure S1A).

Figure S1: Single cell data showing UMAP with sample identification (A), cell type identification (B), CD5L expression in all cell populations (C) and bar graph of CD5L expression identified at single cell level (D).

d. Clustering of the data should be included to visualize how the cell types were detected.

Response: We agree with the reviewer and have now included the clustering of data in the revised Figure S1 (Figure S1B).

2. In figure S1 the authors claim that they “observed almost no expression of CD5L” in any of the six populations including T cells, monocytes, epithelial cells, fibroblasts, natural killer cells, and B cells. Why were these cell types selected? Expression in endothelial cells must be included in this graph for comparison. To exclude that there is another population that is an important source of CD5L, expression in all clusters should be included in the graph.

Response: We can only annotate with high quality these six populations based on our specific markers. We did not detect endothelial cells in these specific samples, as shown below. While there are some PECAM1 positive cells, we do not believe they are real endothelial cells as the other markers are negative and the cluster where we observed some PECAM1 expression is mostly from CD45 positive cells; thus, we cannot confirm they are endothelial cells. To justify the explanation above, we presented the populations representing endothelial cells from this analysis in the following Figure.

Figure: Identification of endothelial cells using canonical markers.

3. Another important point is that the scRNA-seq was performed in human samples. However, in the text the authors use this data to support a selective CD5L expression in endothelial cells in a mouse. A question therefore remains what the expression of CD5L in murine tumor cell types is. Based on Figure S1 it cannot be concluded that other cell types in the murine tumors are irrelevant as a source of CD5L.

Response: Thanks for this comment. We would like to point out that CD5L was identified based on our gene expression profiling data demonstrating CD5L was highly expressed (27.48-fold) in B20-resistant versus B20-sensitive SKOV3ip1 mouse ovarian tumor endothelial cells (as shown in Fig. 1B). Immunohistochemical analysis also supported this observation that CD5L expression in endothelial cells from resistant tumors was higher than in endothelial cells from sensitive tumors (Fig. 1C). The data shown in Fig. S1 was in response to one of the reviewer's questions about CD5L expression in other tumor cell types. While it is formally possible that CD5L may be expressed by some other cell types in murine tumors, it is likely to be a relatively minor contribution.

REVIEWERS' COMMENTS

Reviewer #5 (Remarks to the Author):

General: The authors extended the presentation of single cell RNA-seq data in the manuscript and provided additional data in their response letter. Unfortunately, with more information available it became clear that the experiment does not convincingly show that endothelial cells are the source of CD5L in tumors, or that the other cell types are not the source of CD5L. Because ECs were not detected as a distinct cluster (even though, based on the PECAM1 and VWF UMAPs provided in the response letter, they might be included in the fibroblast cluster), it is in fact impossible to compare CD5L expression in the other cell types, as a positive control is missing. Probably, CD5L is bordering a detection limit of the method and the results shown in Figure S1C are not reliable. If this data is to stay in the manuscript, which in my opinion should be carefully considered, the above-mentioned issue – that the data show negligible expression of CD5L in the tumor environment, but because of the lack of positive control this might represent a technical artifact – must be acknowledged in the manuscript.

Minor comment:

1. The type of graph shown in Supplementary figure 1D is not a “bar plot”, but a “violin plot”, please change the legend accordingly.

Reviewer comments:

Reviewer #5 (Remarks to the Author):

Major comments:

1. The authors extended the presentation of single cell RNA-seq data in the manuscript and provided additional data in their response letter. Unfortunately, with more information available it became clear that the experiment does not convincingly show that endothelial cells are the source of CD5L in tumors, or that the other cell types are not the source of CD5L. Because ECs were not detected as a distinct cluster (even though, based on the PECAM1 and VWF UMAPs provided in the response letter, they might be included in the fibroblast cluster), it is in fact impossible to compare CD5L expression in the other cell types, as a positive control is missing. Probably, CD5L is bordering a detection limit of the method and the results shown in Figure S1C are not reliable. If this data is to stay in the manuscript, which in my opinion should be carefully considered, the above-mentioned issue – that the data show negligible expression of CD5L in the tumor environment, but because of the lack of positive control this might represent a technical artifact – must be acknowledged in the manuscript.

Response: We thank the Reviewer for this comment. We respectfully feel that the single cell data was primarily aimed to show that other cells do not have CD5L expression. The expression of endothelial CD5L has been supported by Figures 1 and 7 included in the manuscript that show CD5L expression in both mouse and human tumor endothelial cells. Nevertheless, we have noted the lack of endothelial cells in the Discussion, as suggested by the reviewer (page 17).

In addition, all the original data from single cell analysis has been deposited to the GEO database with access number GSE181955. We have included the access link:

<https://www.ncbi.nlm.nih.gov/geo/query/acc.cgi?acc=GSE181955>

2. The type of graph shown in Supplementary figure 1D is not a “bar plot”, but a “violin plot”, please change the legend accordingly.

Response: We have revised the legend in Figure S1, as suggested.

Fig. S1. *CD5L* expression in human ovarian tumors.

A total of 5 freshly resected high-grade serous ovarian cancer (HGSC) specimens were collected. Samples were processed using 10X genomic Chromium Single cell 3' v3. UMAP visualization of major cell types were used for subsequent analysis by **(A)** sample and **(B)** cell type. **(C)** UMAP visualization of *CD5L* expression. **(D)** Violin plot of *CD5L* expression in the major cell types. Samples with similar nomenclature such as OMT-1.1 and OMT-1.2 refers to the same patient which were sorted with CD45+ and CD45- before the single cell analysis. The single cell analysis data was deposited in GEO (Accession number GSE181955).